# Long-Term High-Resolution Sediment and Sea Surface Temperature Spatial Patterns in Arctic Nearshore Waters Retrieved Using 30-Year Landsat Archive Imagery

**Konstantin P. Klein** [1,2,*], **Hugues Lantuit** [1,2], **Birgit Heim** [1], **Frank Fell** [3], **David Doxaran** [4] and **Anna M. Irrgang** [1]

1. Department of Permafrost Research, Alfred Wegener Institute Helmholtz Center for Polar and Marine Research, 14473 Potsdam, Germany; hugues.lantuit@awi.de (H.L.); birgit.heim@awi.de (B.H.); anna.irrgang@awi.de (A.M.I.)
2. Institute of Geosciences, University of Potsdam, 14476 Potsdam, Germany
3. Informus GmbH, 13187 Berlin, Germany; fell@informus.de
4. Laboratoire d'Océanographie de Villefranche, UMR 7093 CNRS-SU, 06230 Villefranche-sur-Mer, France; doxaran@obs-vlfr.fr
* Correspondence: konstantin.klein@awi.de

**Abstract:** The Arctic is directly impacted by climate change. The increase in air temperature drives the thawing of permafrost and an increase in coastal erosion and river discharge. This leads to a greater input of sediment and organic matter into coastal waters, which substantially impacts the ecosystems, the subsistence economy of the local population, and the climate because of the transformation of organic matter into greenhouse gases. Yet, the patterns of sediment dispersal in the nearshore zone are not well known, because ships do not often reach shallow waters and satellite remote sensing is traditionally focused on less dynamic environments. The goal of this study is to use the extensive Landsat archive to investigate sediment dispersal patterns specifically on an exemplary Arctic nearshore environment, where field measurements are often scarce. Multiple Landsat scenes were combined to calculate means of sediment dispersal and sea surface temperature under changing seasonal wind conditions in the nearshore zone of Herschel Island Qikiqtaruk in the western Canadian Arctic since 1982. We use observations in the Landsat red and thermal wavebands, as well as a recently published water turbidity algorithm to relate archive wind data to turbidity and sea surface temperature. We map the spatial patterns of turbidity and water temperature at high spatial resolution in order to resolve transport pathways of water and sediment at the water surface. Our results show that these pathways are clearly related to the prevailing wind conditions, being ESE and NW. During easterly wind conditions, both turbidity and water temperature are significantly higher in the nearshore area. The extent of the Mackenzie River plume and coastal erosion are the main explanatory variables for sediment dispersal and sea surface temperature distributions in the study area. During northwesterly wind conditions, the influence of the Mackenzie River plume is negligible. Our results highlight the potential of high spatial resolution Landsat imagery to detect small-scale hydrodynamic processes, but also show the need to specifically tune optical models for Arctic nearshore environments.

**Keywords:** ocean color remote sensing; suspended particulate matter; turbidity; nearshore zone; Herschel Island Qikiqtaruk; river plume; coastal erosion; Landsat

## 1. Introduction

Climate change is stronger in the Arctic than anywhere else on Earth [1]. It leads to multiple impacts on the biophysical system, including intensified permafrost thaw, increasing river discharge and stronger coastal erosion. This, in turn has the potential to mobilize large carbon pools stored in permafrost [2], to release greenhouse gases, such as carbon dioxide ($CO_2$) and methane ($H_4$) to the atmosphere [3,4] and to release carbon and nutrients directly into the nearshore zone [5].

Because of enhanced coastal erosion and increased discharge from rivers, sedimentary inputs to the Arctic nearshore zone and shelf areas rose remarkably in the second half of the 20th century [6–8]. The annual discharge of the Mackenzie River rose by 22% together with an increase of particulate export of 46% from 2003 to 2015 [9,10]. Arctic coasts are also actively eroding leading to an enhanced input of sediment [11]. The mean annual rate of shoreline erosion of Arctic coasts is 0.5 m/a [12]. Higher values up to 10 m/a occur at coasts with high ice content, near deltas of large rivers entering the Arctic Ocean (Mackenzie, Lena, Yenisei, Ob, Kolyma) [12].

Arctic nearshore zones, defined as areas shallower than 20 m, remain under- represented in Arctic oceanography, even though their proportion to the Arctic Ocean surface area is around five times larger compared to the rest of the world's nearshore area to the world's ocean (~7.5% compared to ~1.5%, [5]). This is due to the remote location of the Arctic coasts, the dynamic nature of these environments and the challenging conditions for in situ sensor deployment. However, the nearshore zone plays a crucial role in Arctic biogeochemical cycling and sometimes for local economy. Most of the sediment derived from erosion of coastal permafrost settles in the nearshore zone or gets directly transferred into the greenhouse gases [5,13].

Arctic shelves are the main locus of primary production [14], which is highly dependent on solar light penetration within the water column and thus sensitive to variations of water turbidity [15]. Fresh water input to Arctic shelves is for instance necessary for several species of amphidromous fishes that are central to the subsistence economy of indigenous communities [15,16]. Yet, the exact patterns of sediment dispersal in the Arctic nearshore zone are not well-known and shore-specific processes such as longshore drift or barrier island formation are often not captured by existing datasets.

Remote sensing observations have the potential to overcome these challenges and, at least partly, to monitor variations in sediment release associated with changing climate conditions. Yet, satellite sensors are limited either in spatial or temporal resolution, which means that higher temporal resolution is associated with coarser spatial resolution and vice versa [17]. So far, remote sensing of suspended sediments and organic matter (mainly phytoplankton) in seawater has been based on rather coarse (>250 m) resolution sensors (e.g., SeaWiFs Sea-viewing Wide Field-of-view Sensor, MODIS Moderate-resolution Imaging Spectroradiometer, MERIS Medium Resolution Imaging Spectrometer), which cannot resolve shore-specific fine-scale processes [18]. High spatial resolution sensors onboard Landsat (30m) and Sentinel 2 (10 m) satellite platforms can potentially resolve these processes [18–20]. The extensive Landsat archive, which contains spatially consistent data since 1982, can help to detect multi-year surface changes. Its high spatial resolution is also an asset to resolve small-scale surface current processes in the nearshore zone. However, as Landsat sensors were originally designed for land surface applications, the retrieval of suspended sediment information has been subject to many challenges [21].

In this study, we explore the potential of a new technique where multiple Landsat scenes are stacked to calculate "mean" images for a thirty-year period. These images resolve turbidity and water temperature dispersal patterns at high spatial resolution in the nearshore zone. We applied this technique in a study area around Herschel Island Qikiqtaruk, further on referred to as HIQ, in the western Canadian Arctic under seasonal changing wind conditions. HIQ was selected as a study site because it is impacted by large rates of coastal erosion [22] and is located in close vicinity to the Mackenzie River plume on the Canadian Beaufort Shelf [23]. This allows us to map the sediment dispersal and water temperature spatial patterns related to coastal erosion and freshwater input. The specific objectives are: (i) To characterize small-scale nearshore turbidity and water temperature dynamics and (ii) to identify areas where sediment and organic matter accumulate under the two prevailing wind conditions in

the southern Beaufort Sea (ESE and NW). Landsat imagery from 1982 to 2016 was analyzed for that. We hypothesize that, depending on the wind direction, the nearshore zone of HIQ experiences very limited sediment and freshwater input from the Mackenzie Delta, despite its geographical vicinity.

## 2. Material and Methods

### 2.1. Regional Setting

The focus of this study is on the Beaufort Sea inner shelf waters around HIQ (69°36′ N; 139°04′ W, Figure 1). The study area was chosen because of its close proximity to the Mackenzie Delta, the presence of a strongly eroding coast [22,24,25], and changing hydrodynamics of the Mackenzie River outflow [9]. HIQ is located on the Canadian Beaufort Shelf, which covers less than 2% of the Artic coast shelf area (~64,000 km$^2$) and which is narrow (~100 km) compared to the Eurasian shelves [26,27]. The shelf has a gentle relief up to approximately 80 m water depth, where the shelf break is located. Notable exceptions are the Mackenzie Trough, an up to 300-m deep glacial valley located north east of HIQ, and several smaller underwater valleys with low relief [26].

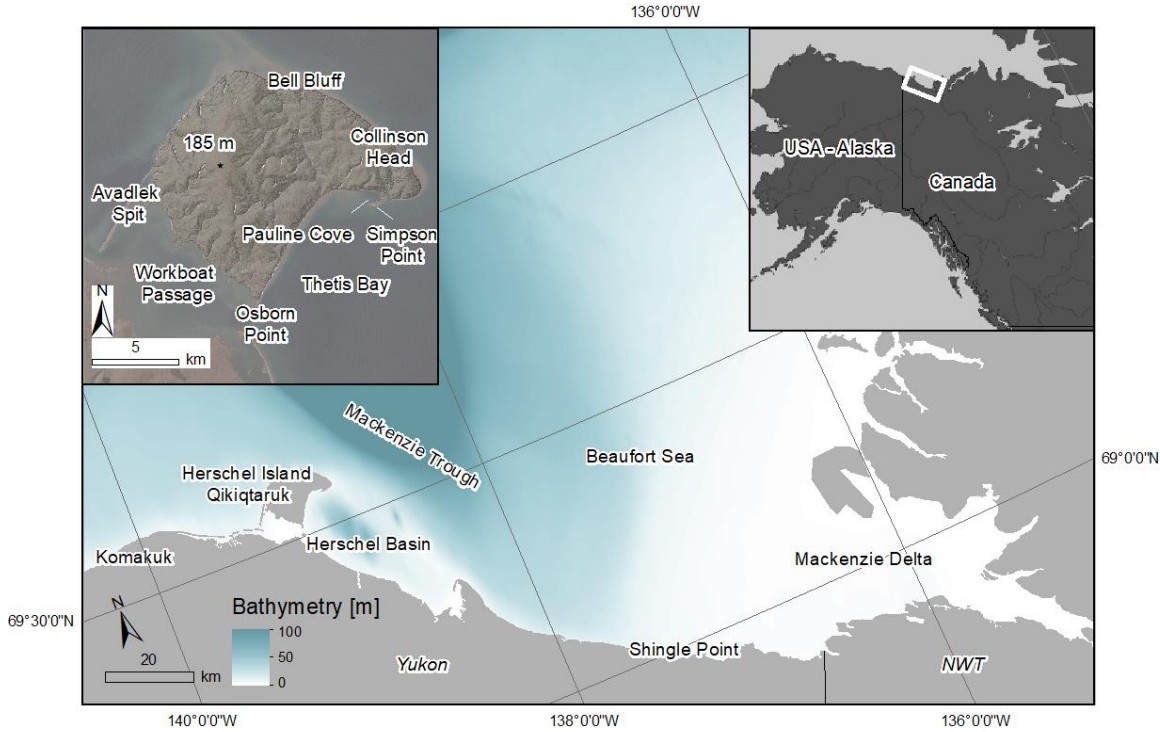

**Figure 1.** Map of the study area. Upper right: Location of the study area (white rectangle) near the northernmost border of Canada and the United States. Main map: Location of the study area in the region of the Yukon Coast, with the Mackenzie Delta in the east, the Yukon Coastal Plain forming the mainland part and HIQ (Herschel Island Qikiqtaruk) in the west. Bathymetry of the Canadian Beaufort Shelf is indicated by different shades of blue (Federal publications Inc: Nautical charts of the Beaufort Sea, 1998–2016). Upper left: close-up to the study area with indicated geographical locations [28]. A Landsat 5 (TM) true color (band composition 321) image is underlain by a hill-shaded 2 m digital surface model.

HIQ is located approximately 120 km west of the Mackenzie River Delta. The Mackenzie River has been identified as the main fresh water and sediment source of the Canadian Beaufort Shelf [23,27,29]. Its ice-free season starts typically in mid-May with a peak discharge in early June (up to 25,000 m$^3$/s) [23,30]. However, it is not unusual to find sea ice at the Canadian Beaufort Shelf until mid-July.

HIQ is separated from the Canadian mainland by the shallow Workboat Passage (<3 m deep, ca. 2 km wide) [31]. Data on water currents are scarce in the area, but maps based on shoreline

morphology indicate that currents predominantly occur in the northern part of Workboat Passage. There, wave action and water currents caused a deepening of the sea floor to its maximum depth of 3 m [31]. The southern part of Workboat Passage is not affected by these currents and is assumed to be a major sediment sink for sediment coming in from the West and from mainland rivers [31]. The north coast of HIQ is erosional and is exposed to maximum wave energy in late summer, while the east coast lies in the lee side of the island and is thus less often affected by storm waves [32,33]. The coastal slopes are affected by excessive thermo-denudation and erosion processes, with several retrogressive-thaw-slumps and active-layer detachment slides [22,24,34].

The climate at HIQ and the southern Beaufort Sea is characterized by long, cold winters and short summers. Monthly mean temperatures vary from about −25 °C in winter (December–February) up to 10 °C during summer (July and August). The mean annual temperature is −9.4 °C (1995–2007) [35,36]. During the open water season, the time of the year when the ocean water is not covered by ice, winds dominantly blow from ESE and NW directions. In August and September, when storms become more frequent, NW wind conditions are more common [33] (Figure 2). Fetch lengths in the Beaufort Sea may extend 1000 km and significant wave heights may exceed 4 m during storms, thereby enhancing the coastal erosion [37]. However, the sedimentary input from coastal erosion to the Beaufort Sea is just approximately 2% of the sedimentary input of the Mackenzie River (1.8 Tg/a compared to 125 Tg/a) [29,33,38].

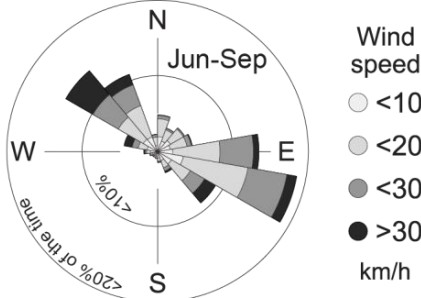

**Figure 2.** Windrose diagram showing wind direction and speed frequency during the sea ice-free period. Data was acquired for 1995–2016 from the weather station at Simpson Point, HIQ. Data provided by [39], figure modified after [38].

## 2.2. Landsat Images Acquisition and Processing

In this work, the following Landsat data products are used:

- Level 1T data in the red and near infrared (NIR) channels are processed through the ACOLITE [18] atmospheric correction scheme to provide the remote sensing reflectance (RRS) and the water turbidity.
- Level 1T data in the thermal infrared are used to derive brightness temperature at the top-of-atmosphere.
- In addition, we use the L2 surface reflectance (RS) product as a means to quality control the ACOLITE-derived remote sensing reflectance (RRS) values.

All Landsat data products were downloaded from the United States Geological Survey (USGS) [40]. The dataset includes images from Thematic Mapper (TM) from 1982 to 2011, Enhanced Thematic Mapper + (ETM+) from 1999 to 2016, and Operational Land Imager/Thermal Infrared Sensor (OLI/TIRS) from 2013 to 2016. Images from ETM+ acquired after 2003 were mostly excluded, because of the failure of the scan line corrector (SLC) leading to data loss in the resulting imagery of ~23% [41]. After 2003, only ETM+ image data from flight path 67 were used in this study, because the study area lies within the central part of the image, where the loss of data is minimal. Since the sea is covered by ice from October to May, only images recorded in June, July, August, and September without significant sea ice and cloud coverage were considered.

The surface reflectance (dimensionless) data product was chosen for this study because it is a standardized atmospherically corrected product processed and provided by the USGS. Landsat TM and ETM+ sensors provide lower radiometric resolution data products compared to Landsat OLI [18]. This results in low signal-to-noise ratios over clear water surfaces, which are darker than land surfaces (except for the blue part of the spectra) or turbid water surfaces. Landsat data products have widely been used for the qualitative retrieval of water column constituents in the past decades [42]. Generally, the water reflectance in the red part of the spectra is well correlated to the light backscattering by suspended sediments in the water column for low to moderate sediment concentrations [43]. Previous studies have demonstrated that RS atmospheric correction algorithms are successfully applied to highly turbid waters [44–47]. However, this has not been tested with Landsat imagery yet.

The atmospheric correction of Landsat satellite data to generate the RS data product was performed by the USGS according to [48]. To assess the performance of the RS (red) data product over water surfaces in contrast to the RRS (red) data product, Landsat scenes processed toward the different reflectance algorithms were compared. These scenes were recorded on September 12th, 2011 by Landsat TM and on August 8th, 2016 by Landsat OLI over the Canadian Beaufort Shelf. Both scenes were downloaded as Level 1T data product without atmospheric correction and as Level 2 data product from the USGS [40]. RRS data products were computed from L1T products using the ACOLITE software (version 20170619.0) [18] and applying the exponential extrapolation algorithm with the two spectral bands in the short-wave infrared (SWIR) part of the spectra as recommended by the authors [20]. The ACOLITE software has been selected because it provides a well-established and documented algorithm, which has been validated and tested in various turbidity settings [49,50]. 1500 point values were extracted from both reflectance products for both Landsat scenes within a wide range of reflectance values using value to point in the spatial analyst toolbox in ArcMap 10.4.1. The plots for comparison were compiled in Python. The comparison reveals that RRS (red) and RS (red) are well correlated for Landsat TM (Figure 3a, $R^2 = 0.95$) and Landsat OLI (Figure 3b, $R^2 = 0.99$). This means that the red wavelength SR data products from both sensors can be used for qualitative investigations of turbidity and SPM. The striping visible in Figure 3a results from the lower radiometric resolution of Landsat TM compared to Landsat OLI, which does not produce binned data. The RS (red) values from both scenes are roughly three times higher (factor $\pi$) than the corresponding RRS (red) value, which is due to that RRS is water-leaving radiance divided by downwelling irradiance and RS is upwelling radiance divided by downwelling radiance.

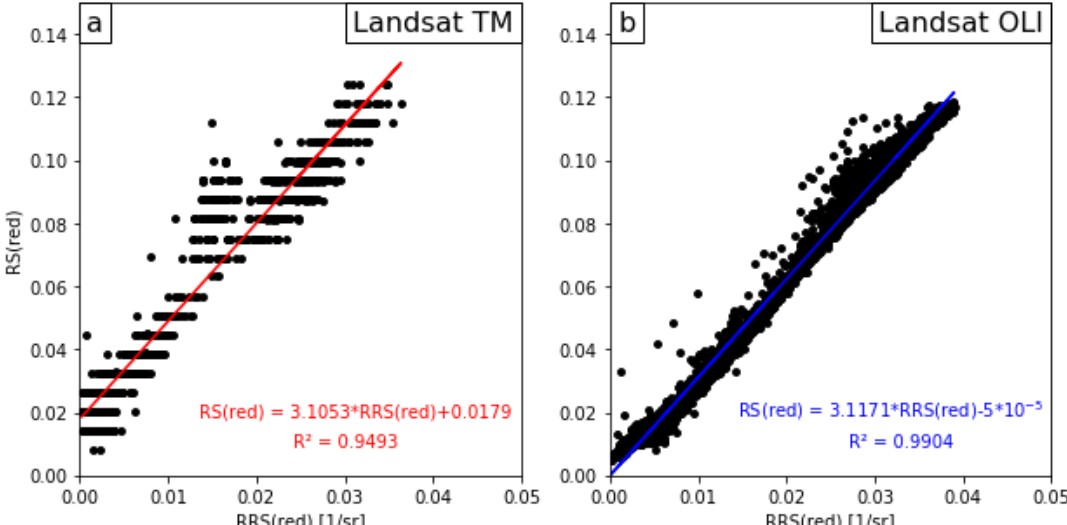

**Figure 3.** Comparison between surface reflectance (RS) (red) and remote sensing reflectance (RRS) (red) data products in coastal and inner shelf waters around HIQ. (**a**) Landsat TM image recorded on September 12th, 2011, and (**b**) Landsat OLI image, recorded on August 8th, 2016.

The RS (red) pre-processed images were classified using the USGS cloud cover and land use detection algorithm "cf-mask" [51] to identify land, sea ice, clouds, and cloud shadow. However, cloud shadows cannot be fully masked as the spectral signature of thin clouds is too close to one of the sediment-loaded waters. The result is a raster image with (almost) only water surfaces (sea, lakes and rivers). A pre-defined area of interest (AOI) around HIQ was used for image processing and analysis to save computing resources. All scenes without artefacts and with matching wind conditions (see below) were used to calculate the mean statistical parameters for visualization.

Landsat TM and ETM+ sensors contain the thermal band, band 6, divided into two, the low gain and high gain bands. The TM and ETM+ thermal band spatial resolutions (original pixel sizes) are respectively 120 m and 60 m, but both products are actually resampled to 30 m pixel resolution. The Landsat 8 platform carries the TIRS with the thermal high gain band 10 and low gain band 11 with 100 m original pixel resolution, also resampled to 30 m. Since TIRS band 10 showed higher accuracy than band 11 [52], the high gain bands were used for all three sensors to assure better comparability of the results. Thermal Infrared image data from all Landsat satellites are not provided with an atmospheric correction by the USGS, thus they are only available as L1T data product. According to [53], the conversion from digital numbers (DN) to spectral radiance ($L_\lambda$) and then to brightness temperature (BT, [K]) is given by

$$L_\lambda \; = \; L_M \times DN + L_A \tag{1}$$

and

$$BT[K] = \; K_2 \; \times \left[ \ln\!\left(\frac{K_1}{L_\lambda} + 1\right)\right]^{-1} \tag{2}$$

with $L_M$ and $L_A$ [W/(m$^2$*sr*μm)] being the radiance multiplier and radiance add, respectively, and $K_1$ [K], $K_2$ [W/(m$^2$*sr*μm)] being thermal constants, which are all given by the Landsat Metafile (MTL). In order to retrieve a temperature in °C as unit,

$$BT[°C] = BT[K] - 237.15 \tag{3}$$

was applied to the results of Equation (2). Hereafter, BT always refers to BT [°C]. All calculations were made with ESRI ArcMap version 10.4.1.

Thirty-five images were used in total for the analysis (TM: 25; ETM+: 8; OLI: 2, Table 1). Several artefacts had to be manually masked from the images before further processing. These anomalies are caused by small sea ice floes or by small clouds and their shadows that were not recognized by the USGS cf-mask.

**Table 1.** List of Landsat scenes used in this study. The data was acquired from the USGS earthexplorer [37]. The wind direction and speed was retrieved from climate data from Environment Canada [49]. The extraction of wind direction is described in Section 2. Changing wind direction refers to unsteady wind conditions.

| Acquisition Date (YYYY-MM-DD) | Sensor | Path/Row | Wind Direction | Mean Wind Speed (km/h) |
|---|---|---|---|---|
| 1986-09-14 | TM | 67/11 | changing | 18.08 |
| 1990-08-17 | TM | 66/11 | ESE | 35.17 |
| 1990-09-16 | TM | 68/11 | NW | 14.50 |
| 1990-09-25 | TM | 67/11 | NW | 17.50 |
| 1992-08-06 | TM | 66/11 | ESE | 27.58 |
| 1992-08-20 | TM | 68/11 | changing | 24.75 |
| 1992-08-29 | TM | 67/11 | changing | 21.33 |
| 1994-07-27 | TM | 66/11 | changing | 11.08 |
| 1994-08-12 | TM | 66/11 | changing | 9.42 |
| 1994-09-11 | TM | 68/11 | ESE | 23.83 |
| 1995-07-12 | TM | 68/11 | ESE | 27.92 |

**Table 1.** *Cont.*

| Acquisition Date (YYYY-MM-DD) | Sensor | Path/Row | Wind Direction | Mean Wind Speed (km/h) |
|---|---|---|---|---|
| 1997-07-19 | TM | 66/11 | ESE | 21.33 |
| 1997-08-02 | TM | 68/11 | NW | 38.92 |
| 1998-07-13 | TM | 67/11 | ESE | 15.00 |
| 1998-07-22 | TM | 66/11 | ESE | 29.08 |
| 1999-08-08 | TM | 68/11 | ESE | 23.00 |
| 1999-08-10 | TM | 66/11 | ESE | 27.42 |
| 1999-09-02 | TM | 67/11 | ESE | 16.75 |
| 1999-09-10 | ETM+ | 67/11 | ESE | 11.00 |
| 1999-09-18 | TM | 67/11 | ESE | 18.33 |
| 1999-09-26 | ETM+ | 67/11 | NW | 38.33 |
| 2001-08-30 | ETM+ | 67/11 | changing | 9.25 |
| 2002-09-11 | ETM+ | 66/11 | NW | 14.42 |
| 2004-08-22 | ETM+ | 67/11 | ESE | 21.42 |
| 2006-07-26 | TM | 68/11 | ESE | 29.33 |
| 2007-08-23 | TM | 67/11 | changing | 23.50 |
| 2008-06-30 | ETM+ | 67/11 | ESE | 21.33 |
| 2009-07-27 | TM | 67/11 | NW | 11.08 |
| 2009-08-21 | TM | 66/11 | changing | 12.17 |
| 2009-09-05 | ETM+ | 67/11 | NW | 10.92 |
| 2010-08-15 | TM | 67/11 | ESE | 19.58 |
| 2011-09-12 | TM | 66/11 | changing | 9.42 |
| 2013-07-15 | OLI | 66/11 | ESE | 18.00 |
| 2014-07-02 | OLI | 66/11 | NW | 16.50 |
| 2016-08-07 | ETM+ | 67/11 | ESE | 24.83 |

Furthermore, the USGS cf-mask did not always properly recognize small or narrow parts of water or land surfaces. This often resulted in false statistical calculations in Pauline Cove and, during ESE wind conditions, west of Avadlek Spit, while Avadlek Spit itself was often not recognized as land surface. These areas have been masked after applying the cf-mask.

All scenes with the same prevailing wind conditions (see below) were stacked together to calculate mean values for each pixel cell.

*2.3. Landsat Turbidity Retrieval*

Since no specifically tuned optical algorithm (i.e., water reflectance vs. water turbidity relationship) for our study area exists, we considered a recently published globally applicable turbidity model designed by [43] and applied it to our dataset. Turbidity (formazin nephelometric unit, FNU) is a well-known and easily measurable proxy for suspended sediment concentration in sediment transport applications [54]. It here refers to a measurement of water clarity, indicating how intensively light is scattered by suspended particles within the water column [55]. The general, semi-analytical turbidity algorithm was calibrated for Landsat TM, ETM+, and OLI imagery using a wide range of field data from coastal and estuarine environments in Europe and South America, based on a switching algorithm of red and NIR RRS values [43]. The algorithm was applied to the Landsat scenes listed in Table 1 using the ACOLITE software [18,20], where turbidity (T) is calculated through:

$$T(red\ or\ NIR) = \frac{A_T^\lambda \rho_W(\lambda)}{(1 - \rho_W(\lambda)/C_T^\lambda)}\ [FNU] \tag{4}$$

and

$$T = (1 - w) \times T(red) + w \times T(NIR) \tag{5}$$

with $\rho_W(\lambda)$ being the water reflectance ($\rho_W = \pi \times RRS$) at wavelength $\lambda$, $A_T$ [FNU] and $C_T$ [dimensionless] being wavelength dependent calibration coefficients, $w$ being a weighing coefficient, *T (red)* being the modelled turbidity using RRS (red), and *T (NIR)* being the turbidity modelled using RRS (NIR) [43]. The RRS (red) formula was used for $\rho_w < 0.05$ and the RRS (NIR) formula was used for $\rho_w > 0.07$, with a linear blending ($w$) in the transition zone $0.05 \leq \rho_w \leq 0.07$ according to Equation (5). The exact retrievals of $A_T$ and $C_T$ is described in [43].

## 2.4. Transects in the Nearshore Zone

In order to investigate the variations in Landsat derived water turbidity close to the coast, a transect of alongshore sampling points was generated starting at Simpson Point and moving clockwise around HIQ (Figure 4). This line consists of points with a spacing of 250 m, resulting in 235 points in total. The distance of each point to the coastline is 75 m, meaning there are two pixels between the USGS-cf mask coastline and the sampling point to avoid mixing pixels of land and water. The value to point function in the spatial analyst toolbox in ArcMap 10.4.1 was used to extract RS (red) and turbidity values at each point of the transect. The extracted points were plotted against each other using Python.

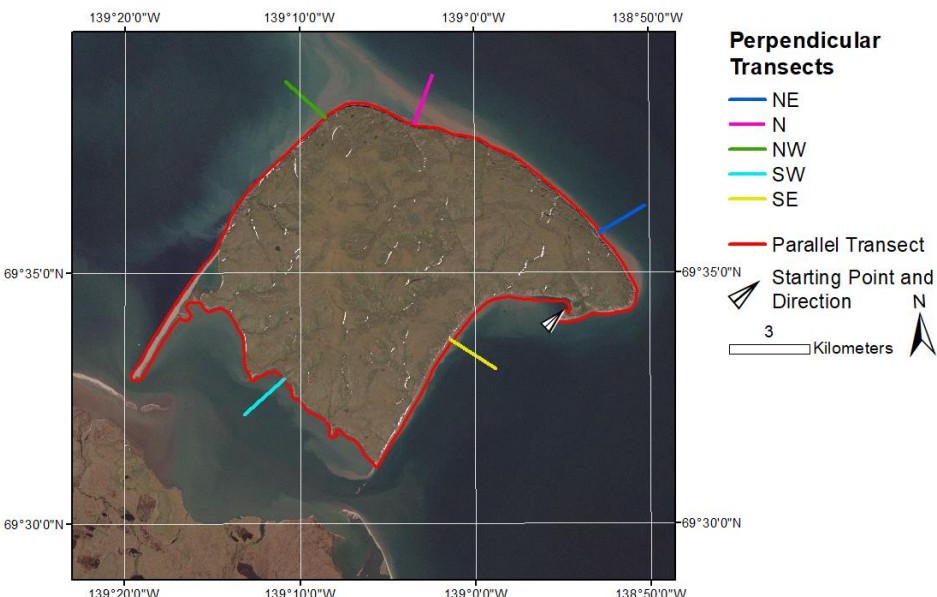

**Figure 4.** Transects in the nearshore zone of HIQ that were used to facilitate the interpretation of gradients of water turbidity, RS (red), and water temperature. The black arrow indicates the starting point and the direction of the alongshore transect. Background image shows a Landsat 5 (TM) true color (band composition 321) image.

In order to facilitate the interpretation of the BT, RS (red) and turbidity images, five transects were designed perpendicular to the coast of HIQ. They cover different coastal orientations and range from 0 m to 2000 m distance to the coastline, with 0 m referring to the first water pixel. Each transect was made of 68 points with a spacing of 30 m to cover every Landsat pixel. The pixel values at each of these points were extracted using the extract multi values to point function in the spatial analyst toolbox in ArcMap 10.4.1. The attribute tables were then exported as .txt and processed using Python.

## 2.5. Wind Data

Wind data were collected for the weather stations HIQ and Komakuk Beach from the climate archive of Environment Canada [39]. Data from Komakuk Beach were used before the HIQ station was set up in 1994 and whenever the weather recording at the HIQ station failed. To receive consistent

data over the whole investigation period, only measurements from 00:00, 06:00, 12:00, and 18:00 were extracted from all datasets, even though hourly data from both stations is available since 1994 [36].

Wind speed and direction observations were acquired for the date of each Landsat scene and the two previous days in order to extract representative dispersal patterns, in total 12 measurements per scene. When six out of the last eight six-hourly measurements (date of the scene plus the previous day) or nine out of twelve had comparable wind directions, the conditions were assumed to be steady and the scene was used for this study.

The wind data was used to reconstruct the general wind direction patterns over several days so that an acquisition of wind data each six hours was sufficient and the only way to provide consistent and comparable data over the whole observation period. Manually controlled random samples in the hourly dataset showed no significant anomalies compared to the six hours interval used per day.

In total, 18 scenes showed continuous ESE wind conditions (TM: 13, ETM+: 4, OLI: 1), 8 continuous NW wind conditions (TM: 4, ETM+: 3, OLI: 1), and 9 showed changing wind conditions (TM: 8, ETM+:1), which refers to unsteady wind conditions during the 12 observations.

## 3. Results

### 3.1. Brightness Temperature

The highest BT values were detected during steady ESE wind conditions along the SE coast of HIQ (>10 °C), and the lowest values near the NW coast (<3 °C, Figure 5c). Changing wind conditions also resulted in high BT at the SE coast as observed during steady ESE wind conditions (~9 °C, Figure 5a). Similar low BT values as the ones detected during steady ESE wind conditions were detected along the NW coast during steady NW wind conditions (~4 °C, Figure 5b). However, both wind regimes (changing wind conditions, and steady NW wind conditions) did not generate a range in BT as large as generated during steady ESE wind conditions (Figure 5c). In short, the highest temperatures were observed during steady ESE wind conditions, except for an area at the NW coast of HIQ, which is shielded from the warm Mackenzie River plume by the island.

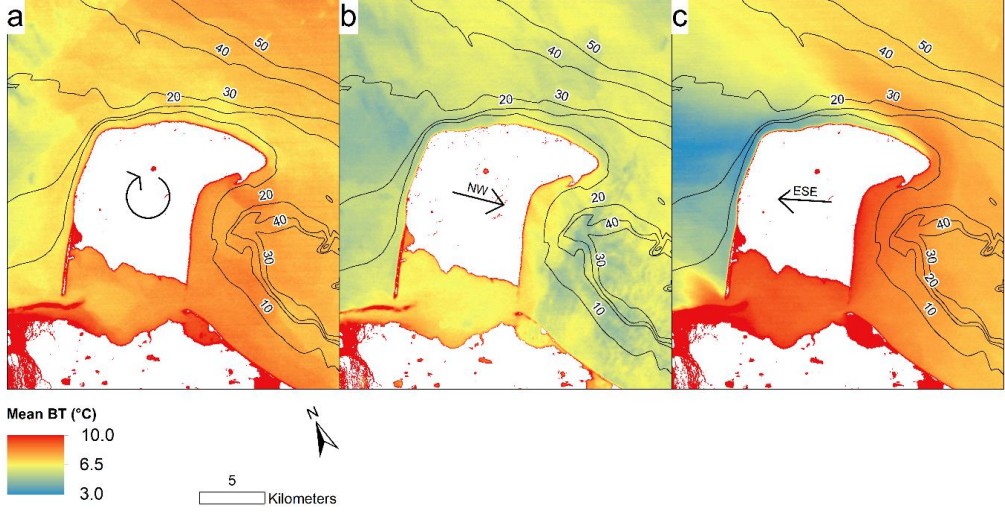

**Figure 5.** Mean BT from Landsat thermal infrared channels, for (**a**) changing wind conditions, (**b**) steady NW wind conditions, and (**c**) steady ESE wind conditions. Prevailing wind conditions are indicated with arrows in the center of each picture. The number of used scenes per wind condition are listed in Table 1. Red areas indicate areas of high BT, white areas indicate land surface areas. Mean BT is highest in (**c**) and lowest in (**b**). Bathymetry is indicated by black lines (Federal publications Inc: Nautical charts of the Beaufort Sea, 1998–2016). Note the very cold BT in (**c**) at the NW coast of HIQ and the large contrast to the SE coast and Workboat Passage. The calculated standard deviation for each image is given in Appendix A.

During steady ESE conditions (Figure 5c), very cold BT values occur at the NW coast of HIQ and there is a large contrast to the SE coast and Workboat Passage. The spatial BT pattern outline that the cold surface water appears in a wake form at the Western part of HIQ. The mean BT values are lowest during steady NW wind conditions (Figure 5b), which is partly caused by a larger number of late summer satellite acquisitions. The influence of bathymetry on BT is limited under all three conditions. The calculated standard deviation for each image is given in Appendix A.

The BT spatial distribution shows similar spatial patterns during changing and steady NW wind conditions, but absolute values are higher during changing wind conditions (~4 °C). During steady NW and changing wind conditions, BT decreases with increasing distance from the coast. This can be seen at the SE and NW coasts, as well as NE of Collinson Head. In contrast under steady ESE wind conditions, BT values rise with increasing distance from the coast along the NE coast.

The Workboat Passage shows comparatively uniform BT for all three wind conditions (Figure 5). The BT between the barrier islands and the Canadian mainland were higher than in the Workboat Passage for all three wind conditions. In the Workboat Passage, the BT were highest during steady ESE wind conditions (~10 °C) and lowest during steady NW wind conditions (~5 °C).

## 3.2. Surface Reflectance and Turbidity Mapping

The mean RS (red) values, which were used as proxy to estimate long-term average spatial patterns in water turbidity, showed the highest values during steady ESE wind conditions (mean reflectance value of 0.0323) and lowest values during steady NW wind conditions (mean value of 0.0241, Figure 6). An area of high turbidity was consistently present at the NE coast of HIQ, an area that is characterized by high cliffs (RS (red) > 0.055). Another area along the SE coast was also characterized consistently by high turbidity (RS (red) > 0.055). This area is characterized by multiple retrogressive thaw slumps. During steady ESE wind conditions, dispersal patterns at the NE coast indicate longshore drift extending toward the West. This results in a broad area of high turbidity at the NW coast, which does not exist during the changing or steady NW wind conditions. In contrast to the temperature dispersal patterns (Figure 5), the reflectance values decreased with increasing distance from the coastline under all three conditions.

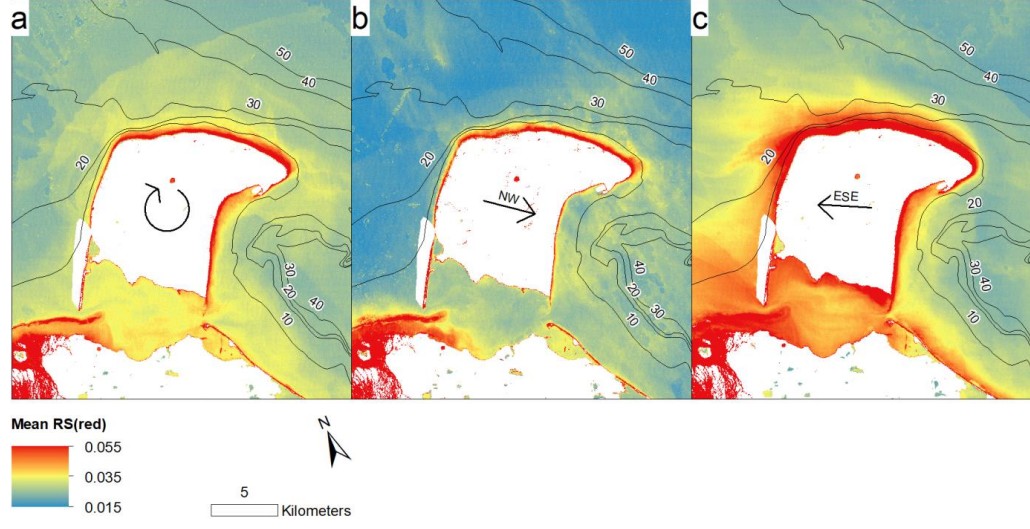

**Figure 6.** Mean RS (red), that was used as proxy for turbidity, for (**a**) changing wind conditions, (**b**) steady NW wind conditions, and (**c**) steady ESE wind conditions. Prevailling wind conditions are indicated with arrows in the center of each picture. The number of used scenes per wind condition are listed in Table 1. Red areas indicate areas of high turbidity, white areas indicate land surfaces or areas of failed atmospheric correction. Bathymetry is indicated by black lines (Federal publications Inc: Nautical charts of the Beaufort Sea, 1998–2016). Turbidity is highest in (**c**) and lowest in (**b**). Note the similar dispersal patterns in (**a**) and (**b**) compared to (**c**). The calculated standard deviation for each image is given in Appendix A.

Bathymetry is a large influencing factor on turbidity, especially along the NE and the NW coast of HIQ. During changing and steady NW wind conditions, high turbidity is limited to shallow water depths (below 10 m, Figure 6a,b). During steady ESE wind conditions, high turbidity extends up to water depths of 30 m along the NE coast and depths of 20 m along the NW coast (Figure 6c). Along the SE coast, high turbidity is limited to the immediate vicinity of the coastline under all three wind conditions.

The Workboat Passage was, under all three conditions, an area of high water turbidity and showed clear sediment pathways, especially close to the inlets (Figure 6c). A thin band of high turbidity (RS (red) > 0.055) could be observed around the whole island. This band showed high turbidity values under all conditions and can be attributed to coastal erosion and resuspension because of waves.

Figure 6a,b, displaying changing wind directions and steady NW winds, respectively, showed rather similar dispersal patterns with a comparable fringe of high turbidity and a strong gradient toward lower values from the coast toward the offshore. Both configurations differ substantially from the sediment dispersal observed under steady ESE winds (Figure 6c).

The Landsat derived turbidity values reveal similar sediment dispersal patterns compared to the RS (red) data, even though the differences between the wind regimes are larger (Figure 7). The gradient from high turbidity in the nearshore (>50 FNU) to lower values offshore (<5 FNU) is larger when applying the [43] turbidity algorithm compared to the RS results (Figure 6). This can be observed along the NW, NE, and along parts of the SE coast of the Island.

During steady ESE wind conditions, a larger filament structure is resolved at the SE coast of the island, where suspended material gets transported toward Herschel Basin (Figure 7c). Several of these filament features were also resolved at the NW coast under all three wind conditions, albeit to a smaller scale. Bathymetry has a comparable influence on turbidity as revealed in Figure 6, especially during steady ESE wind conditions (Figure 7c). During changing and steady NW wind conditions (Figure 7a,b), high turbidity (>50 FNU) is limited to water depths shallower that 10 m.

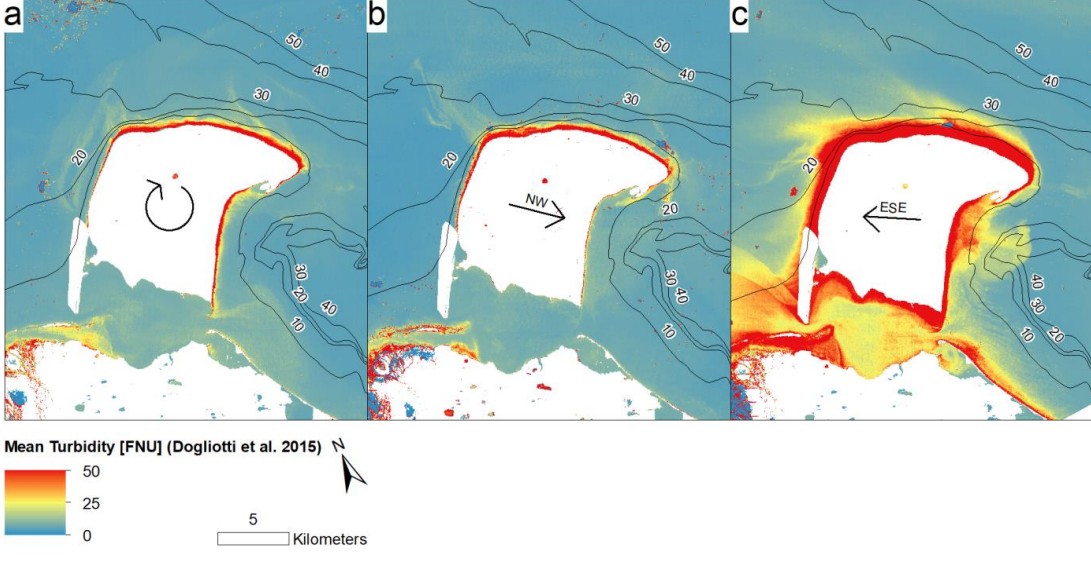

**Figure 7.** Mean Landsat retrieved turbidity after [43] for (**a**) changing wind conditions, (**b**) steady NW wind conditions, and (**c**) steady ESE wind conditions. Prevailing wind conditions are indicated with arrows in the center of each picture. The number of used scenes per wind condition are listed in Table 1. Red areas indicate areas of high turbidity, white areas indicate land surfaces or areas of failed atmospheric correction. Bathymetry is indicated by black lines (Federal publications Inc: Nautical charts of the Beaufort Sea, 1998–2016). Turbidity is highest in (**c**) and similar in (**a**) and (**b**).

*3.3. Gradients in the Nearshore Zone*

Figure 8a shows turbidity and RS (red) values along the coastline of HIQ. In the RS (red) dataset, the highest values were observed in Thetis Bay, along the N coast of the Island, and around Collinson Head (RS (red) ~0.08). This indicates that the sediment supply along the SE coast, which is dominated by retrogressive thaw slumps, is comparable to the one along the NE coast, which is dominated by steep cliffs. Lower values were detected in Pauline Cove and west of Avadlek Spit (RS (red) 0.04–0.05), which are the areas, where the cf-mask did not perform adequately. Thus, these lower values might not reflect reality. The overall variation of RS values is low, except in the two aforementioned regions.

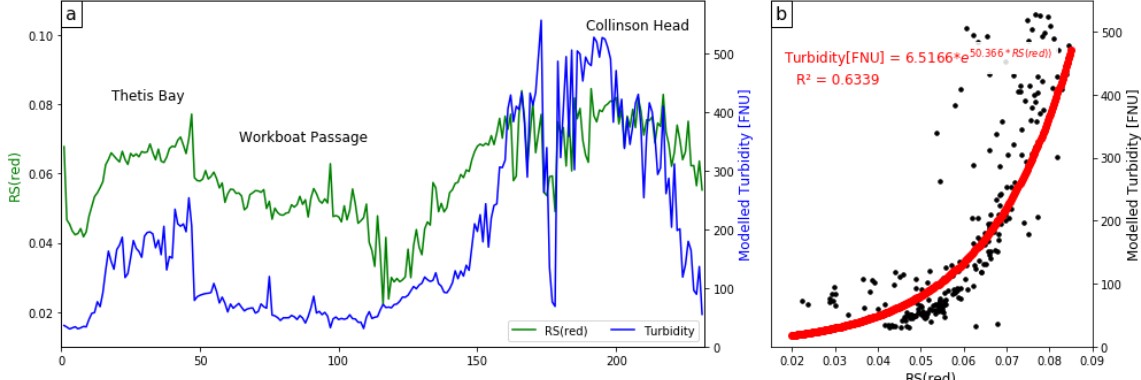

**Figure 8.** (**a**) Comparison between the maximum turbidity and RS (red) values observed along the coast of HIQ. The blue line represents the Dogliotti et al. (2015) [43] turbidity model, the green line the corresponding RS (red) value. (**b**) Correlation between the RS (red) and turbidity values in the nearshore zone. The red line represents the best exponential fitting function. The values were extracted at points along a transect starting at Simpson Point and moving clockwise (Figure 4).

In the turbidity dataset modelled according to [43], the turbidity values along the N coast (up to 500 FNU) are remarkably higher than in Thetis Bay (up to 200 FNU) or in the Workboat Passage (up to 100 FNU). The turbidity values also decrease considerably from 400 FNU at Collinson Head to 50 FNU at Simpson Point. This indicates, in contrast to the RS dataset that the eroding permafrost cliffs along the NE coast provide higher sediment supply to the nearshore zone than retrogressive thaw slumps along the SE coast. The variation of turbidity values around the island is larger compared to the variation of the corresponding RS (red) value, which is due to the exponential design of the turbidity algorithm (Figure 8b).

Figure 8b shows the correlation of the turbidity value with the corresponding RS (red) value displayed in Figure 8a. The correlation of both values is decent, especially for turbidity values below 250 FNU. As previously reported in the literature, the red band saturates at higher turbidity values, which results in a stronger noise [43].

The analysis of the five transects perpendicular to the coastline of HIQ show the aforementioned gradients from the nearshore toward the offshore. Figure 9a,b shows the BT gradients during steady NW and steady ESE wind conditions, respectively. Each transect shows BT decrease within the first 100 m off the coastline, with BT values dropping between 1 °C and 4 °C. The only exception is the NW oriented transect during steady NW wind conditions, which does not show a large variation close to the coastline. During steady ESE wind conditions, the BT values along the NE and N coast rise with increasing distance to the coastline and reach similar values in 2000 m distance as they were direct at the coastline. All other transects do not show a remarkable variation beyond the dropping of BT close to the coast. During steady ESE wind conditions, BT values are remarkably higher at the SE and SW coast than during NW wind conditions (~10 °C compared to ~6 °C). Along the NW coast, the BT are higher during steady NW wind conditions (~5 °C compared to ~3 °C during steady ESE wind conditions). Along the N coast, BT do not differ remarkably during the two wind conditions.

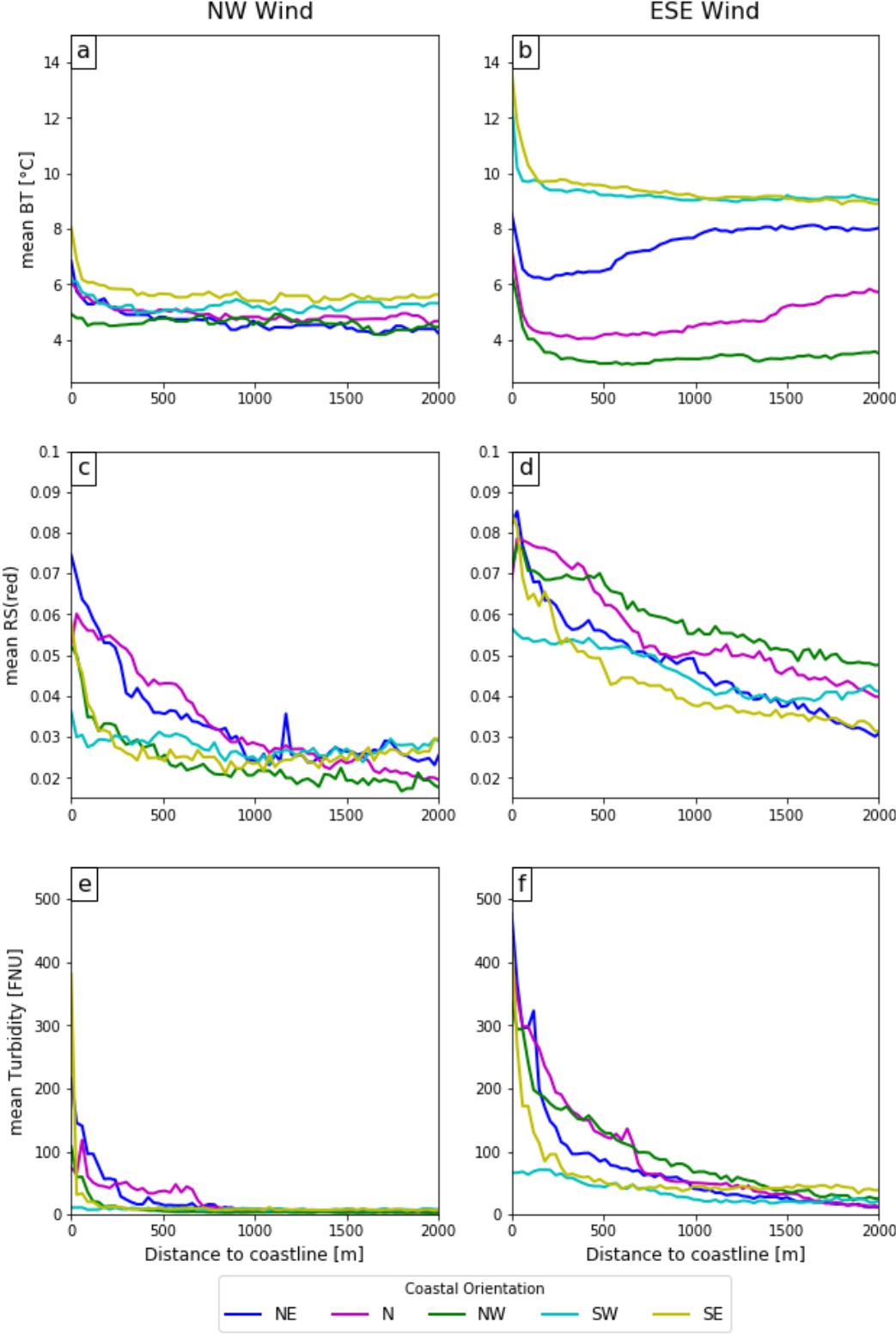

**Figure 9.** Transects perpendicular to the coastline of HIQ to quantify the gradients from the nearshore to the offshore. (**a**) Mean BT during NW wind conditions. (**b**) Mean BT during ESE wind conditions. (**c**) Mean RS (red) during NW wind conditions. (**d**) Mean RS (red) during ESE wind conditions. (**e**) Mean turbidity during NW wind conditions. (**f**) Mean turbidity during ESE wind conditions. Different coastal orientations are indicated by different colors. The position of each transect line is displayed in Figure 4.

Figure 9c,d shows the RS (red) gradients during steady NW and steady ESE wind conditions, respectively. Every transect but one shows a decline of RS (red) with increasing distance from the coastline. The only exception is the SW coast during steady NW wind conditions, where the RS (red) does not change significantly with increasing distance from the coastline. The largest gradient was detected along the NE coast, where the RS (red) value decreases from ~0.075 close to the coast to less than 0.03 in 2000 m distance off the coastline. The RS (red) values during steady ESE wind conditions are higher than the ones during steady NW conditions and roughly twice as much in 2000 m distance off the coast. During steady ESE wind conditions, the RS (red) values do not drop to a background value which does not change significantly with increasing distance off the coastline, while this seem to be the case during steady NW wind conditions at a distance of 1000 m off the coastline (RS (red) ~0.025).

Figure 9e,f show the turbidity gradients during steady NW and steady ESE wind conditions, respectively. Each transect shows a decline of turbidity with increasing distance off the coastline. The only exception is the SW during steady NW wind conditions, where the turbidity does not decline remarkably. In close distance to the coastline, the slope of the declining transect lines is steeper compared to the RS (red) dataset. During steady NW wind conditions, turbidity reaches a background level in 750 m distance off the coast (~5 FNU). During steady ESE wind conditions, turbidity reaches a background level in 1500 m distance off the coast (10 -20 FNU). High turbidity values (>400 FNU) were detected very close to the N, NE, NW, and SE coast during steady ESE wind conditions. During steady NW wind conditions, turbidity values exceed 200 FNU only at the NE and SE coast.

## 4. Discussion

The strong turbidity gradients observed around HIQ from the nearshore to offshore zone demonstrate that Landsat imagery is a powerful instrument to resolve features otherwise not seen in coarser scale imagery as indicated by [18]. This study is one of the first applications of Landsat archive imagery to resolve small-scale sediment dispersal patterns and surface water temperature patterns in Arctic nearshore environments. Figures 8 and 9 reveal high turbidity values, which were not reported in Arctic nearshore environments in the literature yet and highlight the importance of the nearshore zone in the mobilization and transport of sediment and organic matter.

Our results suggest that most of the suspended material in the nearshore zone of HIQ remains in the alongshore surface waters and that offshore transport through surface waters is limited. The material observed close to the coast is likely to be transported alongshore, to quickly settle in the water column or to be exported offshore by bottom currents. The mean RS (red) declines from >0.06 directly at the coast to less than 0.02 within a few hundreds of meters, independently of wind conditions and show the presence of a thin yet sediment-rich band of water along the coast. This highly turbid water, potentially holding large amounts of organic matter, cannot be resolved by the usual applied medium resolution ocean color remote sensing platforms such as MODIS. We show that even the older Landsat sensors (TM, ETM+) are powerful tools for ocean color applications and allow the investigation of Arctic nearshore environments by providing high spatial resolution optical imagery spanning over nearly 35 years.

Our results show that turbidity and RS (red) are well correlated for turbidity values < 250 FNU. This covers most data points in our study area, making the RS (red) a good proxy to map turbidity in Arctic nearshore zones at low to medium sediment loads. However, it is also shown that RS (red) becomes saturated when mapping higher turbidity values, so the turbidity model designed by [43] is most appropriate to map the strong gradients in the nearshore zone. In our images, sediment pathways from the nearshore to the offshore were visible at the NW coast of HIQ and at the SE coast toward Herschel Basin during steady ESE wind conditions. These pathways were not resolved in the averaged RS (red) results which highlights the superior performance of the turbidity model designed by [43] to detect gradients in the nearshore zone. These pathways show that offshore transport of sediment might be observed at the water surface, especially in locations characterized by a large supply of sediment (i.e., retrogressive thaw slums and relatively sheltered shore morphology such as the SE coast of HIQ).

Yet, these features were isolated along the coastline and do not reflect the dominant mechanism for sediment transport under the observed conditions.

Coastal erosion rates are large in the area [22] and contribute large amounts of sediment to the nearshore zone (Figure 10). Figure 10 shows that coastal erosion rates are highest at the NW, N, and NE coasts of HIQ. These coastal sections all showed the highest turbidity in our results, which indicate the supply of significant amount of material to the nearshore zone, albeit in a narrow band located along the coast. Along the SE coast of HIQ, the coastal erosion rates are considerably lower than along the NE coast. Yet, the same relation was detected for turbidity, which supports the hypothesis that coastal erosion is the main contributor of suspended sediment in this area. Generally, the high turbidity gradients observed in our results indicate that most of the eroded material from the coast remains alongshore, rapidly settles at the seafloor or is transported by bottom currents.

Ref. [56] suggests that the distinction between onshore and offshore sedimentation of suspended sediment is a function of wave height. These results indicate that a significant wave height threshold of 1.0 m or above is required to transport suspended sediment offshore. According to [57], significant wave height in the Beaufort Sea exceeds 1.2 m only during strong storm events. In our study area, the majority of waves have a significant wave height of less than 0.8 m [57]. This indicates a limited potential for offshore transport and the potential important role of alongshore transport in mobilizing sediment in this area. Estimations of the amount of suspended sediment transported away from the coast are rare; [58] estimated that approximately 20% of the eroded material by the mainland rivers (Mackenzie River excluded) and coastal erosion is stored in Herschel Basin, a major sedimentary sink at the Canadian Beaufort Shelf. This implies that an important part of the sediments remains close to the coast and contributes to the growth of sedimentary landforms such and spits and barrier islands.

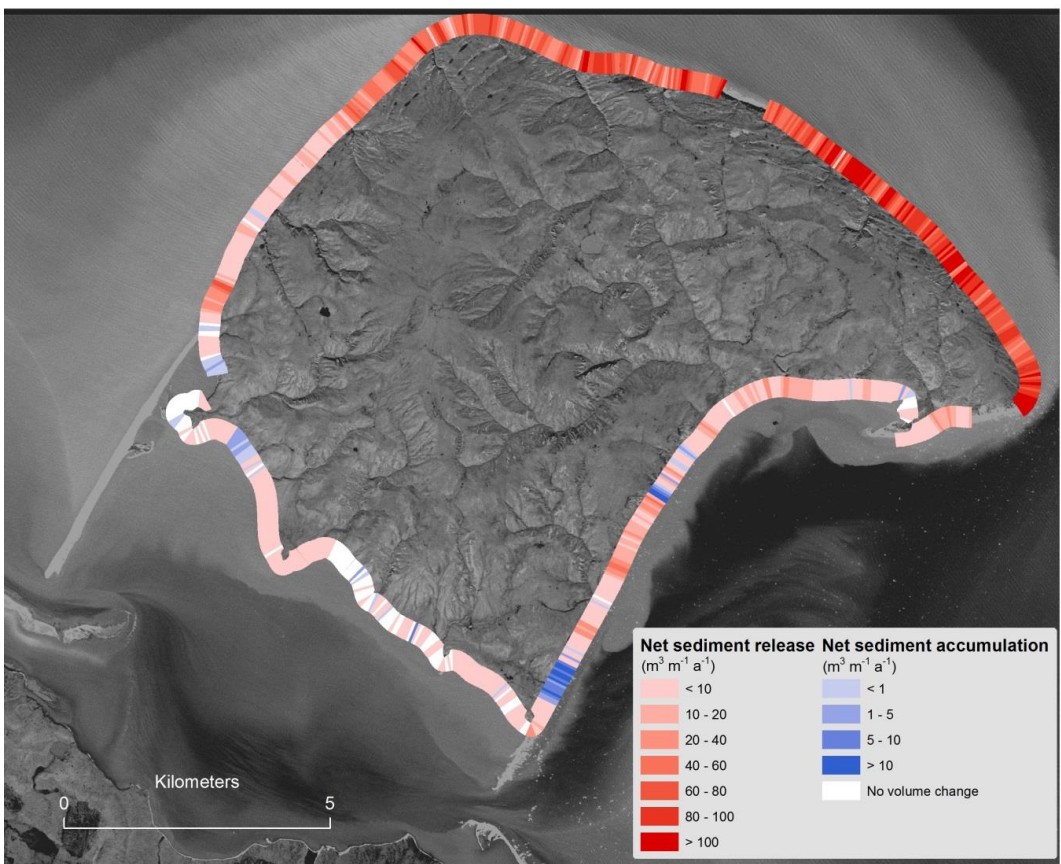

**Figure 10.** Volumetric coastal erosion at HIQ based on DEM elevation changes from 2000-2011 [22].

Riedel (2017) [59] analyzed the origin of the sediment in the western part of Herschel Basin. Using biomarker endmember modeling, the author showed that HIQ is the predominant sedimentary origin at the western edge of Herschel Basin. This shows that offshore transport of sediment is taking place, albeit not at the water surface, a process that is not captured by our satellite images. This also indicates the paramount role of coastal erosion for sediment mobilization from HIQ for the sedimentary budget of Herschel Basin and the adjacent nearshore zone in contrast to the Mackenzie River sedimentary input. The Mackenzie River sedimentary load (125 Tg/a) [29] is much higher than the sediment released by coastal erosion along the Yukon Coast (1.8 Tg/a) [25], yet this load in confined to the immediate vicinity of the delta and does not reach the shore of HIQ in significant amounts. Our results show that the ratio between the sediment amounts exported from coastal erosion and the Mackenzie River is greater during NW wind conditions.

The large turbidity values observed in the nearshore zone (Figures 6–8) are not only associated with fresh material eroded from the coast. Vertical mixing keeps the sediment in suspension. [33] pointed out that bottom currents are driving resuspension in the nearshore zone during storm events, while the influence of waves is higher during normal weather conditions [60]. Ref. [61] detected storm related bottom currents in the Canadian Beaufort Sea three to five times stronger than the mean currents (up to 0.5 m/s). This shows that the high turbidity values observed along the nearshore zone are not necessarily entirely related to coastal erosion, but also to resuspension. It also shows that bottom currents have the potential to contribute to the offshore transport of sediment, which cannot be resolved by our satellite imagery. Another possible trigger for resuspension is upwelling. [62] reported on bottom currents developing beyond the shelf break, moving westwards through the Mackenzie Trough and surfacing NW of HIQ. These currents might add to the high turbidity and low BT values observed during steady ESE wind conditions (Figures 5–7). This also shows that temperature differences are not entirely related to the Mackenzie River plume extent, but also to water masses originating the beyond the shelf break.

Doxaran et al. (2015) [10] suggested an increased occurrence of resuspension because of the rising discharge and sedimentary load of Arctic Rivers at shallow water depths close to their deltas. Assuming that rivers along the Yukon coast experienced similar changes in their discharge regime [63], resuspension along the whole Yukon coast should have been increasing. This is particularly visible in the Workboat Passage. Our data cannot determine whether the resuspension has increased over time, but it supports the fact that the Workboat Passage is characterized by very high turbidity values, which is likely linked to the inputs from creeks and rivers draining into it.

## 5. Conclusions

The aim of this study was to investigate the dynamics of water temperature and turbidity in the coastal and innershelf waters of the Canadian Beaufort Sea around Herschel Island Qikiqtaruk (HIQ). Thirty years of Landsat satellite imagery were analyzed under seasonal changing meteorological forcing. The results showed clear spatial differences for both observed parameters under the two prevailing wind conditions (ESE and NW). During steady ESE wind conditions, turbidity and water temperature values showed significantly higher values in the nearshore zone. The differences in water temperature are mainly caused by the changing extent of the Mackenzie River Plume. Turbidity is mainly driven by local coastal and nearshore processes such as erosion and resuspension. Our results also show that water turbidity is highest close to the north coast of HIQ, which experiences the highest coastal erosion rates, indicating that coastal erosion is the main contributor of sediment to the nearshore zone.

The strong gradient of turbidity from the nearshore to the offshore zone under both steady ESE and steady NW wind conditions indicates that the bulk part of eroded sediment does not remain at the water surface and gets transported alongshore or rapidly settles at the seafloor. The only stable pathway of sediment that get transported to the offshore was identified NNW of HIQ. During steady ESE wind conditions, another pathway was identified SE of the island toward Herschel Basin.

In this study, we showed that Landsat satellite imagery provides coastal observations (e.g., temperature and turbidity within surface waters) at a high spatial resolution in contrast to the coarser spatial resolution of ocean color satellite sensors. Its high spatial resolution and the long time series of spatially consistent data (since 1982) are unique to resolve hydrodynamic processes close to the shoreline and to compare data ranging over more than three decades.

To improve the results of this study and of Landsat based ocean color remote sensing in the area, in situ measurements from the study area are needed to establish an optical model especially tuned to the conditions encountered in the southern Beaufort Sea and the nearshore zone of HIQ. Higher spatial resolution imagery would also enable the resolution of smaller scale sediment dispersal patterns such as resuspension. This could be eventually coupled with geochemical sampling to calculate small-scale sediment and organic matter budget in coastal zones of the Arctic Ocean, which are highly impacted by climate change. Ultimately, our approach bears great potential to resolve processes not covered by other sensors and could be extrapolated to other coastal areas. This will conclusively foster our knowledge about the reaction of Arctic nearshore environments to global climate change.

**Author Contributions:** Conceptualization, K.P.K., H.L. and B.H.; methodology, K.P.K. and D.D.; validation, K.P.K., D.D. and A.M.I.; formal analysis, K.P.K.; writing—original draft preparation, K.P.K.; writing—review and editing, H.L., B.H., F.F., D.D. and A.M.I.; supervision, H.L., B.H., F.F.

**Funding:** This publication is part of the Nunataryuk project. The project has received funding under the European Union's Horizon 2020 Research and Innovation Programme under grant agreement no. 773421. Konstantin P. Klein was financially supported by a Ph.D. stipend by the University of Potsdam (PoGS Potsdam Graduate School).

**Acknowledgments:** USGS and NASA are acknowledged for Landsat imagery. We also would like to thank three anonymous reviewers for their many constructive comments significantly contributing to improve the quality of this manuscript.

**Conflicts of Interest:** The authors declare no conflict of interest.

## Appendix A

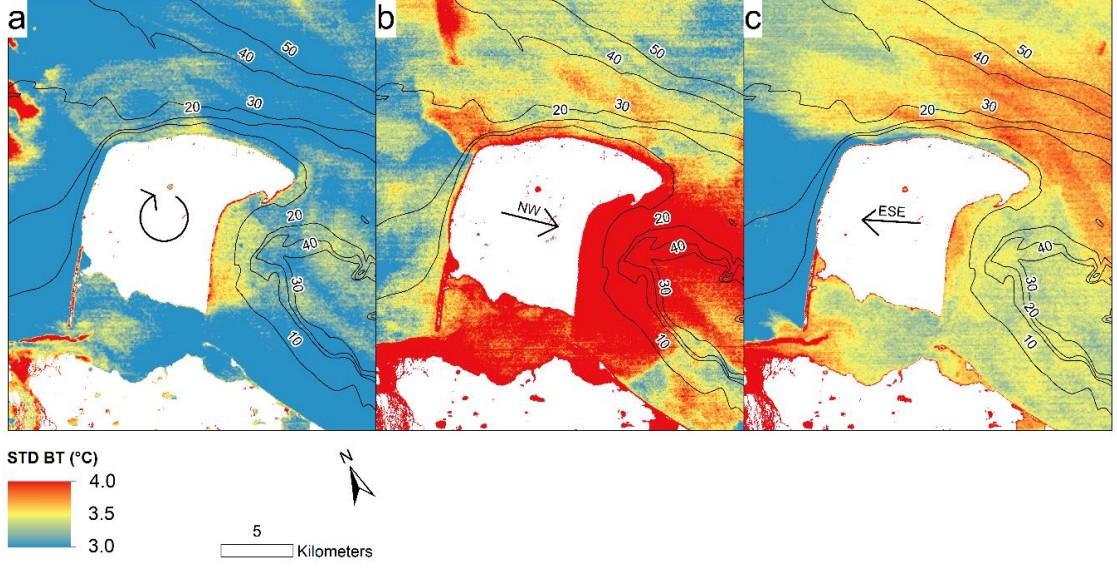

**Figure A1.** Deviation (STD) of BT from thermal infrared channels, for (**a**) changing wind conditions, (**b**) steady NW wind conditions, and (**c**) steady ESE wind conditions. Prevailing wind conditions are mentioned with arrows in the center of each picture. The number of used scenes per wind condition can be seen in Table 1. Red areas indicate areas of high STD, white areas indicate land surface areas.

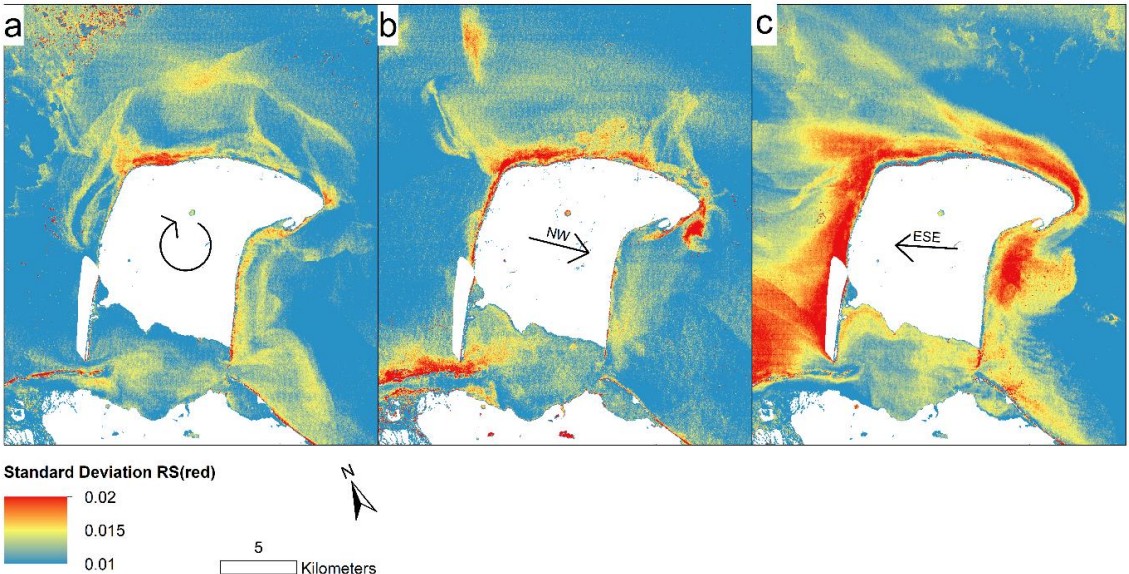

**Figure A2.** Deviation (STD) of RS (red), that was used as proxy for turbidity, for (**a**) changing wind conditions, (**b**) steady NW wind conditions, and (**c**) steady ESE wind conditions. Prevailing wind conditions are mentioned with arrows in the center of each picture. The number of used scenes per wind condition can be seen in Table 1. Red areas indicate areas of high STD, white areas indicate land surfaces or areas of failed atmospheric correction.

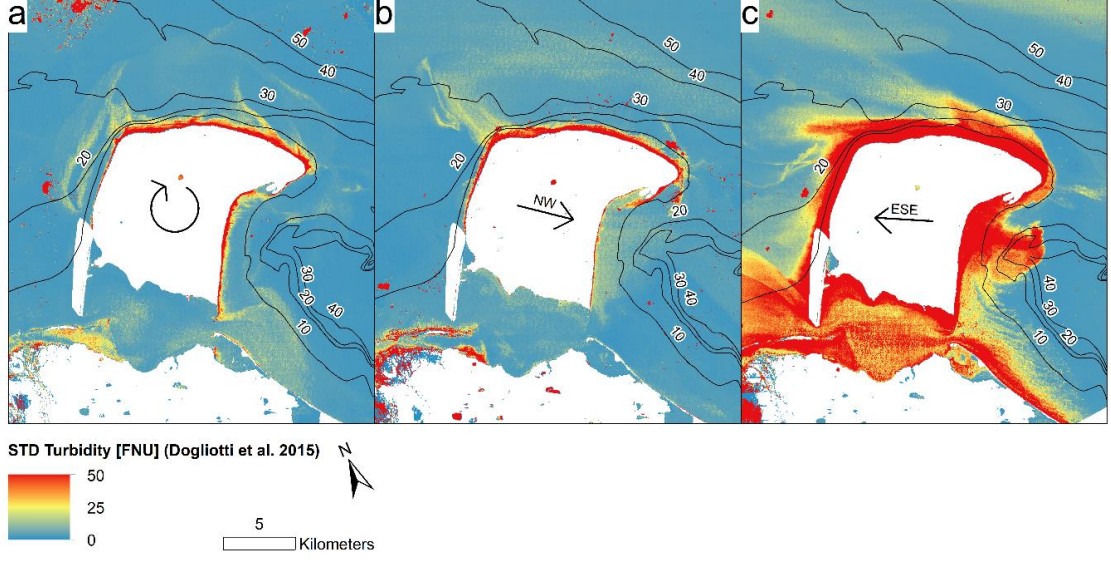

**Figure A3.** Deviation (STD) of turbidity calculated after [43], for (**a**) changing wind conditions, (**b**) steady NW wind conditions, and (**c**) steady ESE wind conditions. Prevailing wind conditions are mentioned with arrows in the center of each picture. The number of used scenes per wind condition can be seen in Table 1. Red areas indicate areas of high STD, white areas indicate land surfaces or areas of failed atmospheric correction.

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
