# Peer review of "Long-Term High-Resolution Sediment and Sea Surface Temperature Spatial Patterns in Arctic Nearshore Waters Retrieved Using 30-Year Landsat Archive Imagery"

_remotesensing, doi:10.3390/rs11232791_

Round 1
Reviewer 1 Report
In this paper, the authors demonstrated a relationship between wind condition with near-shore sea surface temperature and/or turbidity through the case study of sea region surrounding the Herschel Island Qikiqtaruk, Canada. This paper was very improved. Now the study is well designed, the manuscript is well written and organizated and the conclusion is supported and validated.
Therefore, I recommend accept after minor revision. My detailed comments are listed below:
1. The abstract has no explicit the goal. Please add the purpose of the study.
2. The section Introduction is clear written.
3. The section Material and Methods is well written, and the Figures much improved.
4. The section Results is good for interpretation and is written in clear way.
5. The section discussion is very clear written.
6. The Conclusions of a research paper have to summarize the contents and purpose of the study and this is in this paper. But in the sentence "The aim of this study was to investigate the dynamics of water temperature and turbidity in the coastal and innershelf waters of the Canadian Beaufort Sea around HIQ " should be written "The aim of this study was to investigate the dynamics of water temperature and turbidity in the coastal and innershelf waters of the Canadian Beaufort Sea around Herschel Island Qikiqtaruk (HIQ)" .
Later can use the acronym HIQ.
There are also some minor mistakes in the manuscript, e.g. pp 4, l 125 "... [32,33]" sholuld be "...[32, 33]", in the section Reference "Burn, C.R.; Zhang, Y. Permafrost and climate change at Herschel Island ( Qikiqtaruq ), Yukon Territory , Canada. J. Geophys. Res. 2009, 114, 1–16." should be "Burn, C.R.; Zhang, Y. Permafrost and climate change at Herschel Island (Qikiqtaruq), Yukon Territory , Canada. J. Geophys. Res. 2009, 114, 1–16".
Based on this comments above, I would to recommend accept after minor revision.
Author Response
The abstract has no explicit the goal. Please add the purpose of the study.
We have re-phrased the sentence in ll. 19 ff. to explicitly mention the goal of our study in the Abstract.
The Conclusions of a research paper have to summarize the contents and purpose of the study and this is in this paper. But in the sentence "The aim of this study was to investigate the dynamics of water temperature and turbidity in the coastal and innershelf waters of the Canadian Beaufort Sea around HIQ " should be written "The aim of this study was to investigate the dynamics of water temperature and turbidity in the coastal and innershelf waters of the Canadian Beaufort Sea around Herschel Island Qikiqtaruk (HIQ)".
Changed.
There are also some minor mistakes in the manuscript, e.g. pp 4, l 125 "... [32,33]" sholuld be "...[32, 33]"
The references in our manuscript were made according to the giudelines of the journal. As written here (https://www.mdpi.com/journal/remotesensing/instructions), there is no white space supposed to be between two references. We therefore have not changed this in our manuscript.
"Burn, C.R.; Zhang, Y. Permafrost and climate change at Herschel Island ( Qikiqtaruq ), Yukon Territory , Canada. J. Geophys. Res. 2009, 114, 1–16." should be "Burn, C.R.; Zhang, Y. Permafrost and climate change at Herschel Island (Qikiqtaruq), Yukon Territory , Canada. J. Geophys. Res. 2009, 114, 1–16".
Changed.
Reviewer 2 Report
Dear authors,
Congratulations!
Thank you for the time and effort taken to revise the manuscript.
Author Response
Thank you. We have to thank you, too, for your time and effort, which significantly improved the quality of the manuscript.
This manuscript is a resubmission of an earlier submission. The following is a list of the peer review reports and author responses from that submission.
Round 1
Reviewer 1 Report
Review
The paper „Long-Term High Resolution Sediment and Sea Surface Temperature Spatial Patterns in Arctic Nearshore Waters retrieved using 30-year Landsat Archive Imagery" by Konstantin P. Klein, Hugues Lantuit, Birgit Heim, Anna M. Irrgang, Frank Fell, David Doxaran is written in a transparent manner and well-organised.
The aim of this paper was to investigate the dynamics of water temperature and turbidity in the coastal and innershelf waters of the Canadian Beaufort Sea around Herschel Island Qikiqtaruk.
The paper is recommended for publication in Remote Sensing after minor revision.
The main comments:
- The abstract and the goal is clear.
- The interpretation and conclusions are sound and justified by the data.
- The Figure 2 is of bad quality and very fuzzy.
- The Figure 4 it is unreadable and incomprehensible.
- Shortcuts, for example, RS(red) should be RS (red).
- pp15, l453 ...andFigure 11... - should be ...and Figure 11...
- pp19 , l570 ...(Figure 5,Figure 6,Figure 7 andFigure 9) - should be ...(Figure 5, Figure 6, Figure 7 and Figure 9).

Author Response
Comment: “The Figure 2 is of bad quality and very fuzzy.”
Figure 2 has been revised and is now provided in higher quality.
Comment: “The Figure 4 it is unreadable and incomprehensible.”
Figure 4 has been deleted due to the new structure suggested by reviewer 3.
Comment: “- Shortcuts, for example, RS(red) should be RS (red). pp15, l453 ...andFigure 11... - should be ...and Figure 11... pp19 , l570 ...(Figure 5,Figure 6,Figure 7 andFigure 9) - should be ...(Figure 5, Figure 6, Figure 7 and Figure 9).”
Changes were made according to the reviewer’s suggestions.
Reviewer 2 Report
The work described in this manuscript attempts to demonstrate a relationship between wind condition with near-shore sea surface temperature and/or turbidity through the case study of sea region surrounding the Herschel Island Qikiqtaruk, Canada. The manuscript has several flaws in raising research problems/questions and interpreting the obtained result. The authors should clarify several following points:
1) Why are changes in surface turbidity and SST patterns of sea region surrounding the Herschel Island Qikiqtaruk the interests of international communities? Why you aim to describe such a regional case at international publication level (Please check again your research questions and purpose/objectives)
2) Effect of changing wind direction on surface turbidity and SST patterns is obvious. Using satellite data to demonstrate this effect is a traditional remote sensing application (O’Neill et al., 2010; Nehorai et al., 2013), however, should be reviewed in the manuscript to clarify what your new findings are.
O’Neill, L. W., Chelton, D. B., & Esbensen, S. K. (2010). The effects of SST-induced surface wind speed and direction gradients on midlatitude surface vorticity and divergence. Journal of Climate, 23(2), 255-281.
Nehorai, R., Lensky, I. M., Hochman, L., Gertman, I., Brenner, S., Muskin, A., & Lensky, N. G. (2013). Satellite observations of turbidity in the Dead Sea. Journal of Geophysical Research: Oceans, 118(6), 3146-3160.
3) All data using in the manuscript is retrieved from Landsat without any validations of the retrievals. Why did you select ACOLITE to process the Landsat data? Is the turbidity retrieval model (Eq. 5) developed based on Landsat reflectance derived from ACOLITE? If not, how can be sure that the retrieval of turbidity is appropriate?
4) What is “changing wind condition” (lines 300)? Please give a background of all wind conditions.
5) Why were only images acquired in late summer (July to September) selected in this study?
6) Please provide coordination, frame, location information, and legend in all figures. Locations those you described in the result section do not appear on the maps giving difficulty to
7) The trends of mean RS(red) and BT values are nearly similar in the case of NW wind but not similar in the case of ESE wind (Figure 10, 11) that means there is large uncertainty in turbidity pattern under the ESE condition. Please discuss this issue.
8) Please provide objective conclusions to clear out the changes of turbidity and SST patterns under three investigated wind condition (quantitatively).
9) Others:
Line 75: suspended sediment concentration (SSC) - result shows the only turbidity in FNU, that is not SSC
Line 235: "SPM concentration": is SPM different from SSC. Please use a uniform term to state the suspended sediment concentration over the manuscript.
Line 426-427: Is this a sentence?
Author Response
Comment: “Why are changes in surface turbidity and SST patterns of sea region surrounding the Herschel Island Qikiqtaruk the interests of international communities? Why you aim to describe such a regional case at international publication level (Please check again your research questions and purpose/objectives).”
Thank you for this comment. We are aware of the fact that our study describes a small area. Our approach is to explore a new technique (stacking of landsat imagery to map sediment dispersal at high resolution) in the Arctic setting. To do this, we selected an area where we have an intimate knowledge of nearshore sediment dynamics. We have studied Herschel Island Qikiqtaruk since 2003 and use this experience to deploy this new technique which we think is very promising. The technique itself can easily be extrapolated to larger areas and this is precisely what we are working on. The current paper is a first step in that direction and needed to be based on a location we know well. We also think that Herschel Island Qikiqtaruk is a very good first site to deploy this techniques. Herschel Island Qikiqtaruk is impacted by large rates erosion, it is characterized by a diverse geomorphology and bathymetry and is in close vicinity to the Mackenzie Delta which allows us to map sediment dispersal related to coastal erosion AND freshwater input. Finally, in comparison to previously published ocean color studies in the Canadian Beaufort Sea (e.g. Doxaran et al. 2012), we are able to detect gradients in the nearshore zone that were not detected in these studies. We show that nearshore zones are of crucial importance for the mobilization of sediment, nutrients and organic matter that gets eroded from permafrost coasts.
The proportion of nearshore areas to the total Arctic Ocean surface is by far larger than the proportion of nearshore areas to the world’s ocean surface (Fritz et al., 2017). This shows that our results have wide-reaching implications in highlighting the role of the nearshore zone for carbon turnover and ecosystem functions. We have rewritten the introduction section of our manuscript to highlight the relevance of our study to the global scientific community.
Comment: “Effect of changing wind direction on surface turbidity and SST patterns is obvious. Using satellite data to demonstrate this effect is a traditional remote sensing application (O’Neill et al., 2010; Nehorai et al., 2013), however, should be reviewed in the manuscript to clarify what your new findings are.”
The influence of wind induced shear stress on surface water movement and thus turbidity and SST is obvious, indeed. The goal of this study was not to document this relationship, but to highlight the role of other contributing factors in driving sediment dispersal. Our turbidity modelling clearly shows that the nearshore sediment budged has other contributing factors than the Mackenzie River, i.e. coastal erosion and/or resuspension. Previous studies, for example Doxaran et al. 2012, were not able to detect the influence of coastal erosion in the nearshore sediment budged. In short, we show that turbidity and SST react differently on wind forcing and that coastal erosion and/or resuspension mobilize a considerable amount of sediment, but negligible amounts of warm water to the nearshore zone.
Comment: “All data using in the manuscript is retrieved from Landsat without any validations of the retrievals. Why did you select ACOLITE to process the Landsat data? Is the turbidity retrieval model (Eq. 5) developed based on Landsat reflectance derived from ACOLITE? If not, how can be sure that the retrieval of turbidity is appropriate?”
ACOLITE is a well-established, documented algorithm, validated in turbid waters (Vanhellemont 2019). It is a publicly available atmospheric correction tool which can directly be applied to Landsat TM/ETM+/OLI data. We are aware of the existence of other atmospheric correction tools, however, given the exploratory nature of our study, we have purposely chosen to work with well established algorithms. We understand the usefulness of investigating other models, but think that this should be the focus of a new study. The current paper aims to demonstrate the usefulness of the approach in general and suggest improvements for the future, including the use of other algorithms.
Landsat satellite data (the full archive) is also available as surface reflectance products generated by the USGS using ‘land processing standards’ (Masek et al. 2006). Previous studys have demonstrated that this atmospheric correction algorithms are valid for highly turbid waters (Doxaran et al. 2009). We therefore included it into our analysis.
Even though we suggest that ACOLITE is the most appropriate atmospheric correction procedure in our study area, we have compared RRS results calculated with ACOLITE to the RS results downloaded from the USGS to check their consistency. The results obtained are convincing (Fig. 3). Since two different atmospheric correction algorithms provided very consistent results, we assumed that the use of ACOLITE would be the right algorithm to use.
Eq. 5 is a general semi-analytical relationship between the water reflectance and turbidity values. It is appropriate to retrieve the water turbidity in our study area by inversion of atmospherically-corrected Landsat satellite data. The A and C coefficients in Eq. 5 are wavelength-dependent and also depend on the way suspended particles absorb and backscatter light per unit of concentration (mass-specific inherent optical properties). As we do not have in situ data in the study area to locally compute these coefficients, we assume the A and C values computed by Dogliotti et al. (2015) based on a large dataset representative of different coastal and estuarine waters are valid.
Comment: “What is “changing wind condition” (lines 300)? Please give a background of all wind conditions.”
We added this information to the manuscript.
Comment: “Why were only images acquired in late summer (July to September) selected in this study?”
In June, the sea is still covered with sea ice resulting in mixing sea-ice – water pixels. We describe in section 2.2 of the manuscript that only summer acquisitions can be used because the ocean is covered by sea ice for most of the year.
Comment: “Please provide objective conclusions to clear out the changes of turbidity and SST patterns under three investigated wind condition (quantitatively).”
We adapted the conclusion section and added quantitative information.
Other comments: were addressed and/or implemented in the manuscript.
Reviewer 3 Report
It is my opinion that this manuscript needs a MAJOR REVISION. It needs to be shortened, the remaining material made more concise, and arguments made more clearly based on what is actually revealed by the data in this study. My detailed comments are given below.
They authors frequent use of “dispersal pattern” and “pathway” is not justified. Simply observing a particular pattern does not mean you are capturing dispersal, especially as the viewpoint is so localized to the island. And the general implication that the results address “small-scale current processes”, “hydrodynamic processes”, “SST dynamics” is in my view unwarranted. The objectives paragraph in the Introduction should be revised and made clearer.
A small point, but please shorten the annoyingly long “Herschel Island Qikiqtaruk” (perhaps after an initial occurrence?) to simply Herschel Island. "Qikiqtaruk”, as you probably know, is just another word for island.
section 2.2.
I read this several times, but still could not understand it in full. Are you saying the use of normalized radiance in one case vs irradiance in the other is a source of noise? Is Eq. 1 and discussion included because this is a step the authors perform themselves? If so, how is RS actually calculated? If not, why include these details at all?
Table 1. Why not also provide the mean wind speed? Will than confirm — as shown in the wind rose diagram — that ESE events have higher speed? Also, use of “unstable” or “stable” can be confused for atmospheric (gravitational) stability. Better to use steady or changing winds. Perhaps I missed this, but does “changing” mean ALL the useable imagery that does not meet criterion NW and ESE?
Several of the claims made by the authors need more careful consideration.
An example is the claim made on lines 370-371: “The sediment transport pathways which are well resolved in the RS dataset in the Workboat Passage are not resolved in 370 Figure 7 (c).” First, I am not sure what pathways, if any, are captured; second, to my eyes, the RS and turbidity patterns in the Passage are essentially identical. Interestingly, turbidity appears to show more detail than RS. An example is Fig. 7c, which shows on the east side of the island what seems to be a sort of “ejection event”, that is, the sort of yellow “mushroom and stem” shape. A very similar sediment structure in about the same location can be seen in other types of high-resolution imagery; e.g., WorldView03 imagery from 09July2019. The fact that such a feature appears in the MEAN field of Fig. 7c seems noteworthy, as it does indeed suggest a true pathway.
I don’t understand why turbidity values in Fig. 8 extend beyond 500 units, while in Fig. 7 the scale goes only to 50.
section 3.2
This section is tedious, and I really am none the wiser after going through it. I recommend deleting the ROI analysis.
See also below.
section 3.2 (again), but now “Temporal Trends
This material should be deleted. There can be no expectation of detecting trends with such a limited dataset, especially as the only “control variable” is wind direction.
Discussion section
One expects some summary discussion of which, in the author’s opinion, provides the better indicator of suspended sediment, RS or turbidity? Also, why is the sediment distribution generally broader under ESE?
The reader should not have to wait until Figure 12 to see bathymetric contours! The relationship of the signals to bathymetry is a key result and should be shown immediately. Contours should be shown on all the early relevant figures, and Fig. 12 should be deleted.
The authors mention the strong offshore gradient many times, but they quantify it only on line 501. It might help if a representative offshore profile or two were shown directly in a new figure, both for RS and turbidity, and superimpose the local bathymetry.
The authors might add a new figure such as Fig. 7 in their reference number 23, “Fifty years of coastal erosion and retrogressive thaw slump activity on Herschel Island, southern Beaufort Sea, Yukon Territory, Canada”. Such a direct comparison with measured coastal retreat rate would seem to support the authors interpretations.
The fact that there is a river discharging into the Passage should be mentioned earlier!
I find the discussion of upwelling (lines 567 …) to be rather implausible.
This is a problem with showing such a localized view, while the really important dispersal pathways are of much larger scale. For example, I examined MODIS/Terra color and SST imagery from 6-7Aug2016. (I chose this date simply because the final LANDSAT image listed in Table 1 is from 7Aug2016.) Basically, the SST evolution suggests to me that the cold water on the west side of the island under ESE conditions is a local “wake like” effect; if there’s upwelling, it is likely to be localized.
line 583, “Only small parts of it …” This is another example where the authors overstate what the analysis can possibly reveal. Material does not accumulate indefinitely along the shore, so of course the material is transported offshore. You simply are not capturing those pathways, either because of signal-to-noise issues or sampling constraints. That such pathways are apparently NOT revealed in the mean RS or turbidity is a key result and should be emphasized, as you do here — but discuss the caveats, too.
line 587. This is not a conclusion; it is a given fact.
Finally, I think the number of references is excessive, given the limited scope of the findings.
Author Response
Comment: “They authors frequent use of “dispersal pattern” and “pathway” is not justified. Simply observing a particular pattern does not mean you are capturing dispersal, especially as the viewpoint is so localized to the island. And the general implication that the results address “small-scale current processes”, “hydrodynamic processes”, “SST dynamics” is in my view unwarranted. The objectives paragraph in the Introduction should be revised and made clearer.”
Thank you for this comment. We have discussed this question before we initially submitted the manuscript. We decided to use the term ‘dispersal’ because we observe several patterns not only once, but, in case of ESE wind directions, 16 times. The fact that these features, for example the longshore drift along the NE coast of the island or the fine lamination in the Workboat Passage (between the island and the mainland), are still visible after calculating the mean of each pixel cell over 16 Landsat scenes, made us interpreting them as dispersal patterns and not as single events. An alternative solution would be “regular” or “frequent” patterns but given the remarkable patterns obtained for the 30 years of observation, we decided to use the term “dispersal”.
The objective paragraph has been revised.
Comment: “A small point, but please shorten the annoyingly long “Herschel Island Qikiqtaruk” (perhaps after an initial occurrence?) to simply Herschel Island. "Qikiqtaruk”, as you probably know, is just another word for island.”
We introduced the acronym HIQ to describe the study area. Herschel Island Qikiqtaruk is the official name of the territorial park and we therefore decided to not use the term Herschel Island only. Qikiqtaruk is, as described by Burn and Zhang (2009), the name given by the Inuvialuit to Herschel Island and is not just another name for ‘island’ in this area.
Comment: “I read this several times, but still could not understand it in full. Are you saying the use of normalized radiance in one case vs irradiance in the other is a source of noise? Is Eq. 1 and discussion included because this is a step the authors perform themselves? If so, how is RS actually calculated? If not, why include these details at all?”
We re-structured section 2.2 and introduced all remote sensing products and their corresponding processing level. We additionally clarified which products were used for which analysis. The normalized radiance in one case vs irradiance in the other is not a source of noise. This sentence led to misunderstandings and was therefore removed.
Comment: “Table 1. Why not also provide the mean wind speed? Will than confirm — as shown in the wind rose diagram — that ESE events have higher speed? Also, use of “unstable” or “stable” can be confused for atmospheric (gravitational) stability. Better to use steady or changing winds. Perhaps I missed this, but does “changing” mean ALL the useable imagery that does not meet criterion NW and ESE?”
Thank you for this very good comment. We have added the mean wind speed to table 1 and changed the terms in the manuscript according to the comment. We have also added a description of the “changing wind” pattern to the methods.
Comment: “Several of the claims made by the authors need more careful consideration.
An example is the claim made on lines 370-371: “The sediment transport pathways which are well resolved in the RS dataset in the Workboat Passage are not resolved in 370 Figure 7 (c).” First, I am not sure what pathways, if any, are captured; second, to my eyes, the RS and turbidity patterns in the Passage are essentially identical. Interestingly, turbidity appears to show more detail than RS. An example is Fig. 7c, which shows on the east side of the island what seems to be a sort of “ejection event”, that is, the sort of yellow “mushroom and stem” shape. A very similar sediment structure in about the same location can be seen in other types of high-resolution imagery; e.g., WorldView03 imagery from 09July2019. The fact that such a feature appears in the MEAN field of Fig. 7c seems noteworthy, as it does indeed suggest a true pathway.”
This is a very good point. We have added a detailed description of the ‘mushroom’ structure to the results section and discussed it in a later section.
In our interpretation, the fine lamination of yellow (medium turbid) and red (highly turbid) in the Workboat Passage (figure 5 c) shows pathways of water and thus sediment towards the west. This laminated structure is not resolved in figure 7 c, which is noteworthy in our opinion. The general patterns are, as you stated in the comment, similar, with high turbidity close to coast of Herschel Island Qikiqtaruk, lower turbidity between the island and the mainland, and high turbidity behind the barrier islands.
Comment: “I don’t understand why turbidity values in Fig. 8 extend beyond 500 units, while in Fig. 7 the scale goes only to 50."
We decided to use this scale to provide better comparability to figure 5. Another scaling of this figure would result in a smaller red area (high turbidity) and a larger blue area (low turbidity), but would not provide new insights in dispersal patterns due to the large gradients. We therefore decided to use other figures to point out the very high values in direct vicinity to the coastline.
Comment: “section 3.2 This section is tedious, and I really am none the wiser after going through it. I recommend deleting the ROI analysis. See also below.” And “The authors mention the strong offshore gradient many times, but they quantify it only on line 501. It might help if a representative offshore profile or two were shown directly in a new figure, both for RS and turbidity, and superimpose the local bathymetry.”
Thanks for this very good comment. We understand that the gradients can be observed through other means than the ROIs provided in this analysis. We have tried to use transects drawn from the coast to the offshore to show these gradients, but realized that we want to show more than the gradients along these transects. This analysis was defined to perform this transect analysis (ROIs aligned orthogonally to the shore) but also to provide the means to compare ROIs and settings in different areas of the island. We also think that the ROIs are useful because they summarize well the spatial variability observed in nearshore waters. We have followed the reviewer’s suggestions, however, and have produced a revised version of the figure showing the transects. These are more intuitive to read, as pointed out by the reviewer and should facilitate interepretation.The ROI analysis was removed from the manuscript
Comment: “section 3.2 (again), but now “Temporal Trends. This material should be deleted. There can be no expectation of detecting trends with such a limited dataset, especially as the only “control variable” is wind direction.”
We are aware that our dataset is limited to perform such an analysis. During the conception of this paper, we anticipated that such an analysis would be requested by the reviewers and decided to add it to the manuscript, acknowledging its limits. We decided to follow the suggestions made by the reviewer and deleted this analysis.
Comment: “One expects some summary discussion of which, in the author’s opinion, provides the better indicator of suspended sediment, RS or turbidity? Also, why is the sediment distribution generally broader under ESE?”
We added this information to the discussion section.
Comment: “The reader should not have to wait until Figure 12 to see bathymetric contours! The relationship of the signals to bathymetry is a key result and should be shown immediately. Contours should be shown on all the early relevant figures, and Fig. 12 should be deleted.”
We added bathymetric contours to all our figures and address it now in the results section. Figure 12 is deleted.
Comment: “The authors might add a new figure such as Fig. 7 in their reference number 23, “Fifty years of coastal erosion and retrogressive thaw slump activity on Herschel Island, southern Beaufort Sea, Yukon Territory, Canada”. Such a direct comparison with measured coastal retreat rate would seem to support the authors interpretations.
Very good comment. We added a similar, but more recent figure (Obu et al., 2017) to the revised manuscript.
Comment: “The fact that there is a river discharging into the Passage should be mentioned earlier!”
Changed according to the reviewers comment.
Comment: “I find the discussion of upwelling (lines 567 …) to be rather implausible.
This is a problem with showing such a localized view, while the really important dispersal pathways are of much larger scale. For example, I examined MODIS/Terra color and SST imagery from 6-7Aug2016. (I chose this date simply because the final LANDSAT image listed in Table 1 is from 7Aug2016.) Basically, the SST evolution suggests to me that the cold water on the west side of the island under ESE conditions is a local “wake like” effect; if there’s upwelling, it is likely to be localized.”
We are aware of the fact that upwelling acts on much larger scales than presented in this study. However, we interpret the results of Williams et al. (2006) in the way that water, which passes the Mackenzie Trough, moves westward on the shelf. This interpretation (and the wording) is consistent with analyses performed in the area by regional experts (Williams and Carmack: “Ocean Water and Sea Ice”, in “Herschel Island Qikiqtaryuk”, Christopher R. Burn (ed), 2012) and recent modeling results (Machuca 2019, https://open.library.ubc.ca/cIRcle/collections/ubctheses/24/items/1.0378375). We therefore think that this a plausible explanation for the dispersal patterns we have observed and noteworthy in the discussion section.
We modified our discussion part slightly and mention both interpretations in the revised manuscript.
Comment: “line 583, “Only small parts of it …” This is another example where the authors overstate what the analysis can possibly reveal. Material does not accumulate indefinitely along the shore, so of course the material is transported offshore. You simply are not capturing those pathways, either because of signal-to-noise issues or sampling constraints. That such pathways are apparently NOT revealed in the mean RS or turbidity is a key result and should be emphasized, as you do here — but discuss the caveats, too.”
Thanks for this very good comment. We added this to the discussion section. Besides that, we did not state that eroded material would indefinitely accumulate in the nearshore area. We are aware that we cannot finally investigate this with our dataset. We provided our interpretation of the gradients we detected and combined this information with modeling results (Bailard 1982) and oceanographic observations from the Canadian Beaufort Shelf (Hill and Nadeau 1989). However, we understand that the wording we use might have overstated the results of the analysis and modified the manuscript accordingly.
Round 2
Reviewer 2 Report
Dear authors,
My questions/comment stated out your manuscript’s weaknesses. Therefore, you should revise your manuscript consistently to every comment that helps you improve your paper quality. Do not just answer the reviewer. We would like to see your answers on the revised version of the manuscript.
Comment: “Why are changes in surface turbidity and SST patterns of sea region surrounding the Herschel Island Qikiqtaruk the interests of international communities? Why you aim to describe such a regional case at international publication level (Please check again your research questions and purpose/objectives).”
Thank you for this comment. We are aware of the fact that our study describes a small area. Our approach is to explore a new technique (stacking of landsat imagery to map sediment dispersal at high resolution) in the Arctic setting. To do this, we selected an area where we have an intimate knowledge of nearshore sediment dynamics. We have studied Herschel Island Qikiqtaruk since 2003 and use this experience to deploy this new technique which we think is very promising. The technique itself can easily be extrapolated to larger areas and this is precisely what we are working on. The current paper is a first step in that direction and needed to be based on a location we know well. We also think that Herschel Island Qikiqtaruk is a very good first site to deploy this techniques. Herschel Island Qikiqtaruk is impacted by large rates erosion, it is characterized by a diverse geomorphology and bathymetry and is in close vicinity to the Mackenzie Delta which allows us to map sediment dispersal related to coastal erosion AND freshwater input. Finally, in comparison to previously published ocean color studies in the Canadian Beaufort Sea (e.g. Doxaran et al. 2012), we are able to detect gradients in the nearshore zone that were not detected in these studies. We show that nearshore zones are of crucial importance for the mobilization of sediment, nutrients and organic matter that gets eroded from permafrost coasts.
The proportion of nearshore areas to the total Arctic Ocean surface is by far larger than the proportion of nearshore areas to the world’s ocean surface (Fritz et al., 2017). This shows that our results have wide-reaching implications in highlighting the role of the nearshore zone for carbon turnover and ecosystem functions. We have rewritten the introduction section of our manuscript to highlight the relevance of our study to the global scientific community.
Q: Please show me your revision in which line? which page? that clear out this mention.
Comment: “Effect of changing wind direction on surface turbidity and SST patterns is obvious. Using satellite data to demonstrate this effect is a traditional remote sensing application (O’Neill et al., 2010; Nehorai et al., 2013), however, should be reviewed in the manuscript to clarify what your new findings are.”
The influence of wind induced shear stress on surface water movement and thus turbidity and SST is obvious, indeed. The goal of this study was not to document this relationship, but to highlight the role of other contributing factors in driving sediment dispersal. Our turbidity modelling clearly shows that the nearshore sediment budged has other contributing factors than the Mackenzie River, i.e. coastal erosion and/or resuspension. Previous studies, for example Doxaran et al. 2012, were not able to detect the influence of coastal erosion in the nearshore sediment budged. In short, we show that turbidity and SST react differently on wind forcing and that coastal erosion and/or resuspension mobilize a considerable amount of sediment, but negligible amounts of warm water to the nearshore zone.
Comment: “All data using in the manuscript is retrieved from Landsat without any validations of the retrievals. Why did you select ACOLITE to process the Landsat data? Is the turbidity retrieval model (Eq. 5) developed based on Landsat reflectance derived from ACOLITE? If not, how can be sure that the retrieval of turbidity is appropriate?”
ACOLITE is a well-established, documented algorithm, validated in turbid waters (Vanhellemont 2019). It is a publicly available atmospheric correction tool which can directly be applied to Landsat TM/ETM+/OLI data. We are aware of the existence of other atmospheric correction tools, however, given the exploratory nature of our study, we have purposely chosen to work with well established algorithms. We understand the usefulness of investigating other models, but think that this should be the focus of a new study. The current paper aims to demonstrate the usefulness of the approach in general and suggest improvements for the future, including the use of other algorithms.
Landsat satellite data (the full archive) is also available as surface reflectance products generated by the USGS using ‘land processing standards’ (Masek et al. 2006). Previous studys have demonstrated that this atmospheric correction algorithms are valid for highly turbid waters (Doxaran et al. 2009). We therefore included it into our analysis.
Even though we suggest that ACOLITE is the most appropriate atmospheric correction procedure in our study area, we have compared RRS results calculated with ACOLITE to the RS results downloaded from the USGS to check their consistency. The results obtained are convincing (Fig. 3). Since two different atmospheric correction algorithms provided very consistent results, we assumed that the use of ACOLITE would be the right algorithm to use.
Eq. 5 is a general semi-analytical relationship between the water reflectance and turbidity values. It is appropriate to retrieve the water turbidity in our study area by inversion of atmospherically-corrected Landsat satellite data. The A and C coefficients in Eq. 5 are wavelength-dependent and also depend on the way suspended particles absorb and backscatter light per unit of concentration (mass-specific inherent optical properties). As we do not have in situ data in the study area to locally compute these coefficients, we assume the A and C values computed by Dogliotti et al. (2015) based on a large dataset representative of different coastal and estuarine waters are valid.
Q: Please add this explanation in your methodology. Using only a single band (the red band) without validation does not provide a good lesson for the turbidity estimation using Landsat data. Please cite more work that proved ACOLITE is appropriate to estimate turbidity using only red band.
“Previous studys have demonstrated that this atmospheric correction algorithms are valid for highly turbid waters (Doxaran et al. 2009)”. Please recheck the “Doxaran et al. 2009”, at that time there was neither L8 data nor ACOLITE algorithm for L8.
Comment: “What is “changing wind condition” (lines 300)? Please give a background of all wind conditions.”
We added this information to the manuscript.
Q: in which line?
Comment: “Why were only images acquired in late summer (July to September) selected in this study?”
In June, the sea is still covered with sea ice resulting in mixing sea-ice – water pixels. We describe in section 2.2 of the manuscript that only summer acquisitions can be used because the ocean is covered by sea ice for most of the year.
OK
Comment: “Please provide objective conclusions to clear out the changes of turbidity and SST patterns under three investigated wind condition (quantitatively).”
We adapted the conclusion section and added quantitative information.
Q: Please re-check the conclusion and objectives. Is the conclusion provides the results for your study objectives?
Other comments: were addressed and/or implemented in the manuscript.
Q: Should state clearly here your revisions (which line? Which page?)
Legend in Figure 4 is unreadable. Land areas in all figure are not similar. The second figure in the Appendix section, part a show the BT on the land area?Reviewer 3 Report
I am happy to see that a number of my suggestions have been attended to; as a result, the manuscript is leaner and the main points are somewhat easier to discern. I am, however, sorry to say that the manuscript cannot be accepted in its present form. The present version contains errors (“typos”), apparent inconsistencies, and incomplete actions to remedy items in original review. The Discussion section in particular needs improvement. Below are detailed comments that I think should help the authors in preparing a manuscript that better conveys their main findings. I think the scope of the revisions goes beyond "minor", but not necessarily as far as "major". As I must check one of the choices, I have checked the major revision option.
Abstract
l. 22 “and medians” should be deleted, as none are shown in this version of the paper.
This is likely a remnant from the original version, but it should have been corrected.
l. 26 You do not “map the transport pathways”; you infer and conjecture as to what these might be from the patterns of SPM and BT.
l. 29-32 (for example) First state what YOU find, then what that result might be consistent with.
l. 34 and Conclusions. Why is a “bio” component needed in this particular case? This needs justification in a later sections.
Section 2
l. 113. “major sediment sink”? Is this truly what you want to say here? Certainly the island’s coast bounding this area is not one of net accumulation — your Fig. 10. Do you mean that the area receives a lot of particulate material but that currents then transport it away, so no net accumulation; hence, not really a sink?
Table 1. There are still misspellings in the column headings
Fig. 4. Needs improvement.
What is the background image? Why is it so faint?
Difficult to make out all transects; labeling is generally too small.
North arrow appears to not be parallel to the longitude lines.
Section 3
l. 314 To match the order of presentation elsewhere, the calculated standard deviation for BT should appear first in the appendix; it now does not.
l. 316-318 The author's explanation elsewhere in the paper is simpler: the western part is simply shielded from the warm Mackenzie plume.
l. 324 rewrite as: BT decreases
l. 340-342. Incorrect use of language.
l. 364 ff; and your reply to my comment about Fig. 7c vs 6c. See your “laminations” paragraph in your response letter.
Within the “fringe” I think we are seeing filaments of suspended material that is being draw away from the nearshore zone. I still claim to see the same filaments in both RS and turbidity. If the authors disagree, then perhaps they should label in the figure those laminations they see only in RS.
l. 371
In the author’s response letter they claim to “have added a detailed description of the “mushroom” structure to the results section”, as well as discussing it further later on in the paper.
I do not think that is true. All I see is the following very brief sentence:
“During steady ESE wind conditions, a small plume structure is resolved at the SE coast of the island towards Herschel Basin (Figure 7 c).”
Furthermore, the use of “plume” here is a poor choice of word, given that it is used the describe the discharge from the Mackenzie River. I do like the use of “island” vs HIQ.
l. 401-403 This material belongs in the Discussion
l. 431. Suggest rewriting as “Every transect but one shows …”
Discussion section
This section needs to be revised so that the main points become clear and unambiguous, and questionable material is removed so as not to detract from what needs and should be written.
Some sample problem areas:
l. 460-461. How do the authors know this?! Do they have a sediment transport model? A radiative transfer model? Have they considered the size fraction, and how quickly material may be settling into the water column? Water-column stratification? The “optic depth” of the sensor wavelength bands being used?
l. 473-4. Repeatedly, the authors are making apparently contradictory, or at least conflicting, statements such as this one.
If there are no offshore pathways, then what are the “filaments”; what is the “mushroom”?
l. 474 I see no evidence of a “frontal system”. Delete, or explain what you mean.
l. 476-478. There is no evidence of frontal instability in the data. Delete this.
l. 480-481. Seriously? Coastal erosion occurs close to the shore? Who would have expected that?
l. 485-486 Ambiguous, as written. And what does “vastly settles” mean?
Conclusion section
Again, please insure that what is said here is consistent with the rest of the paper, including the Abstract.
In particular, the “discovered indicators” should be explicitly enumerated in the Discussion.
Appendix
Why is there no calculated standard deviation for turbidity? It should either be shown, or the authors need to explain why they choose not to present it. In light of the arguments above concerning the significance of certain offshore-directed pathways, the turbidity s.d. seems relevant to the paper and should therefore be included.